

# 1 Saturated and unsaturated salt transport in peat from a constructed
# 2 fen

Reuven B. Simhayov[1], Tobias K. D. Weber[1,2], Jonathan S. Price[1]
[1]Department of Geography, University of Waterloo, Waterloo, Ontario, N2L 3G1, Canada
[2]Soil Science and Soil Physics Division, Institute of Geoecology, TU Braunschweig, Langer Kamp 19c, 38106
Braunschweig, Germany
*Correspondence to*: Reuven B. Simhayov (rbsimhay@uwaterloo.ca)
**Abstract.** To determine the underlying processes of solute transport in peat an experimental constructed fen peatland, soil
hydraulic properties were measured and saturated and unsaturated solute breakthrough experiments were performed using
$Na^+$ and $Cl^-$ as reactive and non-reactive solutes, respectively. We tested the performance of three solute transport models,
including the classical equilibrium Convection-Dispersion Equation (CDE), a chemical non equilibrium one-site adsorption
model (OSA) and a model to account for physical non-equilibrium, the mobile-immobile phases (MIM). The selection was
motivated by the fact that the applicability of the MIM in peat soils finds a wide consensus. However, results from inverse
modelling and a robust statistical evaluation of this peat provide evidence that the measured breakthrough of the
conservative tracer, $Cl^-$ could be simulated well using the CDE. This is demonstrated by a very high Damköhler number
(→infinity) suggesting instantaneous equilibration between the mobile and immobile phases; this underscores the
redundancy of the MIM approach for this particular peat. Scanning electron microscope images of the peat show the typical
multi-pore size distributions structure have been homogenised sufficiently by decomposition, such that physical non-
equilibrium solute transport no longer governs the transport process. This is corroborated by the fact the soil hydraulic
properties were adequately described using a unimodal van Genuchten-Mualem model between saturation and a pressure
head of ~ -1000 cm of water. Hence, MIM is not the most suitable choice, and the long tailing of the $Na^+$ breakthrough curve
is caused by chemical non-equilibrium. Successful description was possible using the OSA model. To test our results for the
unsaturated case, we conducted an unsaturated steady state evaporation experiment to drive $Na^+$ and $Cl^-$ transport. Using the
parameterised transport models from the saturated experiments, we could numerically simulate the unsaturated transport
using Hydrus-1D. The simulation showed a good prediction of observed values, confirming the suitability of the parameters
for use in a slightly unsaturated transport simulation. The findings improve the understanding of solute redistribution in the
constructed fen.
Keywords: solute transport, peat, non-equilibrium, unsaturated,



## 1 Introduction

The incorporation of large quantities of tailings sand into a constructed fen watershed created as part of a novel attempt at landscape reclamation in the oil sands region, introduced a large pool of leachable Na, Ca and S (Simhayov et al., 2017). Near surface accumulation and potential impact of these solutes on the vegetation is controlled by the transport rate from the upland to the fen and the rate of flushing out of the system, which are currently under investigation (Simhayov et al., 2017). In constructed peatlands designed for oil sands reclamation landscapes, water quality is a concern due to incorporation of process-affected materials (Price et al., 2011; Daly et al., 2012). In this context a better understanding of the transport processes through peat and solute accumulation in the rooting zone of the fen is needed.

The current assumption is that solute attenuation in peat is a result of mass exchanges between mobile and immobile phases (Hoag and Price, 1997; Rezanezhad et al., 2012). Generally, *Sphagnum* derived peat, hyaline cells and their skeletal remnants are thought to account for a large fraction of dead end pores with distinct pore size density distributions (Weber et al., 2017a, 2017b) and a volumetric moisture content (VMC) between 10 and 20% (Hayward and Clymo, 1982; Weber et al., 2017a; 2017b). Additionally, surface adsorption of reactive solutes (Rezanezhad et al., 2012; 2016) may be present. Both effects lead to more dilute but longer solute plumes which might affect revegetation efforts in oil sands reclamation landscapes. The physical and hydraulic properties of undisturbed peat changes along a continuous vertical profile (Weber et al. 2017b, Limpens et al. 2008), whereby deep peat layers are generally more decomposed (Clymo, 1983). In addition to pore-scale effects, the systematic layered heterogeneity common in natural peatlands influences mixing and transport (Hoag and Price, 1995). However, in constructed peatlands this is destroyed because of the disruption caused by stripping, transport and placement (Nwaishi et al., 2015).

Solute transport in the subsurface may be subject to physical and chemical non-equilibrium (Nielsen et al., 1986) invalidating the use of the conventional convection dispersion equation (CDE) to simulate it. Physical non-equilibrium is thought to be a process of a heterogeneous flow field with spatial differences in hydraulic conductivity due to dead-end pores (Coats and Smith, 1964, Zurmühl and Durner, 1996), non-moving intra-aggregate water (Philip, 1968; Passioura, 1971), or stagnant water in thin liquid films around soil particles (Nielsen et al. 1986). In this mobile-immobile model (MIM, Coats and Smith, 1964; van Genuchten and Wierenga, 1976) the liquid phase is partitioned into a mobile and an immobile region. Convective-dispersive transport occurs only in the mobile zone, while solute transport into the immobile region is by diffusion, the rate of which can be determined by experiments and inverse estimation of transport parameters (Vanderborght et al., 1997). In chemical non-equilibrium models, it is assumed that sorption at the pore-water solid particle interface is kinetically controlled (Cameron and Klute, 1977; Nkedi-Kizza et al. 1989). Both parametric non-equilibrium models may additionally account for chemical equilibrium adsorption (Toride et al. 1993).

To distinguish between the governing solute transport process, inverse modelling can provide the necessary information on model parameter estimates, associated uncertainties, and permits the calculation of model performance and selection criteria



(Iden and Durner, 2008; Vrugt and Dane, 2005, Weber et al. 2017). This can be based on measured solute breakthrough
experiments of reactive as well as non-reactive solutes (Nkedi-Kizza et al., 1984). In the notation of the convection
dispersion equation the retardation factor is strictly referred and attributed to equilibrium adsorption (Toride et al., 1995,
Šimůnek and van Genuchten, 2008, Šimůnek et al. 2008) and is a function of bulk density, the slope of the adsorption
isotherm, and volumetric water content (Toride et al., 1995). A problem in deriving a numerical value for the retardation
factor during inverse modelling is that it is mathematical directly negatively proportional to the flow.
To date the only literature reports with experiments of NaCl breakthrough on saturated peat columns conducted in the
laboratory are from Price and Woo (1988), Ours et al. (1997), Hoag and Price (1997) and Rezanezhad et al. (2012). Ours et
al. (1997) speculate that the observed prolonged tailing of NaCl is a result of solutes diffusing into immobile zones.
However, neither batch adsorption tests with the potential to exclude kinetic chemical sorption are presented, nor were solute
transport models fitted to breakthrough curves, leaving their conclusions tentative. Hoag and Price (1997) successfully
described their observations with the conventional convection dispersion equation (CDE). However, based on an effective
porosity ($n_e$) determined by photo imagery the authors calculated the calculate pore water velocity by $v = q/n_e$, where $q$ is
specific discharge. This results in higher values of $v$ than those calculated from total porosity, $\varphi$, or inverse estimation. By
estimating CDE model parameters describing non-reactive Cl$^-$ breakthrough and keeping $v$ fixed, their retardation factor ($R$),
reflecting $v_{water}/v_{solute}$ was >1, and close to the ratio of $\varphi$ to $n_e$. They attributed the delay in solute transport to physical
non-equilibrium processes, whereby solutes diffuse into inactive pores (i.e. solute transfer from the mobile to the immobile
region). The approach of Hoag and Price (1997) differs from the classical understanding where diffusion into the immobile
zone is described by a kinetic constant, while $R$ assumes chemical equilibrium of solutes with sorption sites (e.g. Coats and
Smith, 1964; van Genuchten and Wierenga, 1976). Rezanezhad et al. (2012) concluded MIM transport is observable on
small peat samples Moreover, parameter uncertainties and correlations are not shown and the performance of the MIM in
comparison to the classical CDE is not given such that a rigorous model selection is not possible.
The goal of the study is to expand our understanding of the transport processes in the vadose zone of decomposed peat by
testing various transport models and scrutinizing the common assumption that the mobile-immobile transport model best
reflects the processes in saturated and unsaturated peat. We approach this by conducting saturated breakthrough experiments
using NaCl. Cl$^-$ is generally uninvolved in chemical reactions in peat, except for ultra-saline conditions (Ours et al., 1997),
and its counter-ion, Na$^+$, is a prominent solute and potential contaminant in the oil sands reclamation landscape (Simhayov et
al., 2017). To compare the performance of models, model parameters were estimated using inverse modelling with the
CXTFIT v2.0 code (Toride et al. 1995). Comparison was based on a statistical analysis to investigate the information content
of the data collected, enabling a careful assessment of the underlying processes. Subsequently, the parameterised models
were used to numerically simulate the solute transport in unsaturated steady state evaporation experiments with Hydrus-1D
(Simunek et al., 2008). We tested if the model selection and parameterization based on saturated experiments can be
extended to predict unsaturated solute transport. No further inverse estimation was done for the unsaturated transport of the
non-reactive solute except for the Freundlich-Langmuir parameters of the reactive solute. A sensitivity analyses was then





carried out to estimate potential errors caused by using parameters derived from saturated transport to simulate the
unsaturated case.
**2 Materials and methods**
The peat used for the fen was moderately decomposed rich fen, sedge peat, taken from a donor fen prior to stripping the
overburden material to expose the oil sands deposits (Price et al., 2011; Daly et al., 2012; Nwaishi et al., 2015). Peat samples
were taken from a donor fen prior to stripping the overburden material to expose the oil sands deposits (Price et al., 2011;
Daly et al., 2012; Nwaishi et al., 2015). The donor fen had been drained for two years prior to stripping and the peat
underwent rapid decomposition due to exposure to oxygen (Nwaishi et al., 2015). The peat has a relatively open structure
(Fig. 1), compared to *Sphagnum* peat used in other transport studies (e.g. Hoag and Price, 1989; Rezanezhad et al., 2012).

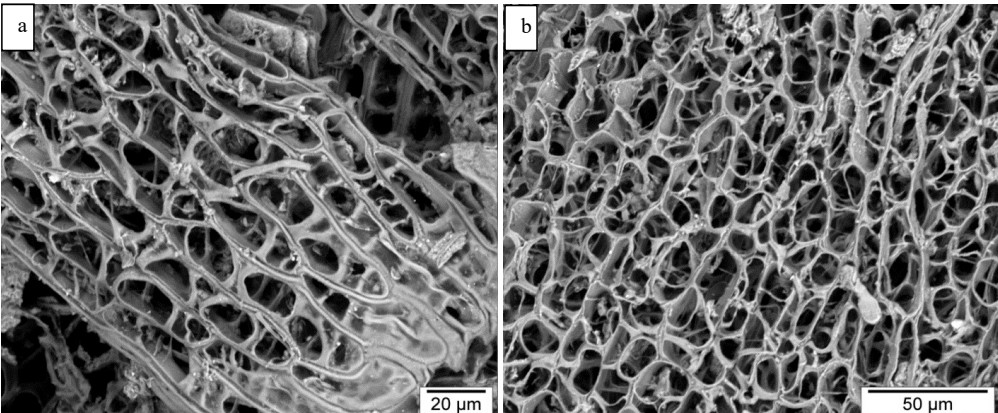


**Figure 1 – Scanning electron microscope pictures of samples of the peat used in this study. a) Moss with hyaline cells,**
**note cells with intact membrane at bottom right corner and larger pore spaces in bottom left and top right corners. b)**
**Moss cells, Note that membranes are missing and a view through the skeleton is evident. Modified from Rezanezhad**
**et al., (2016)**
**2.1 Research approach**
Four soil physical experiments were conducted to estimate the hydraulic properties and solute transport characteristics of the
fen peat material. The conducted experiments were: 1) Transient evaporation; 2) Water retention characteristics using
tension disks; 3) Saturated breakthrough; and 4) Unsaturated breakthrough. Initially, the peat was cleaned from woody plant





material such as leaves and stems, to increase replicability. Furthermore, to reduce variations in moisture content between
samples, the peat was thoroughly mixed before a sample was packed (see appendix A.1). Prior to experimentation samples
were saturated from the bottom in small increments, over a 24-hour period, using 18.2MΩ·cm water (ultra-pure water). All
experiments were conducted at a target bulk density, ($\rho_B$; g cm$^{-3}$), of 0.12. Experiments were conducted in triplicates, except
for the tension disk experiments, which were conducted on 4 samples. For the determination of the solute transport
properties, Cl$^-$ was used as a non-reactive solute and Na$^+$ as a reactive solute. All breakthrough experiments were performed
using a solution of 200 mg l$^{-1}$ Na$^+$ and 300 mg l$^{-1}$ Cl$^-$ corresponding to values measured by Kessel (2016) in the constructed
Nikanotee Fen watershed. This solution was prepared by mixing 500 mg of NaCl (1.06404.055, ACS grade, Merck,
Germany) per 1 l of ultra-pure water.
**2.2 Soil hydraulic properties**
To determine the peat soil hydraulic properties, we conducted transient evaporation experiments (Schindler, 1980, Peters et
al., 2015, Weber et al., 2017a, 2017b) for the retention properties, supplemented with tension disk experiments (Klute and
Dirksen, 1986; McCarter et al., 2017). Tension disk experiments are considered to be a more reliable method to determine
the unsaturated hydraulic conductivity in the wet range, because transient evaporation experiments contain limited
information at pressures heads between 0 and -60 cm for the unsaturated hydraulic conductivity curve (Peters and Durner,
2008). Water retention and unsaturated hydraulic conductivity data were obtained with the tension disk experiments using 10
cm i.d. and 5 cm high peat samples at -2.5, -5, -7.5, -10, -15, -20, -25 cm pressure head (h; cm) steps, which was also the
order in which the experiment was conducted. Outflow during each pressure step was monitored by scales with an accuracy
of 0.1 g and logged at 1-minute intervals and the unsaturated hydraulic conductivity calculated with the Darcy-Buckingham
equation (Swartzendruber, 1969). The transient evaporation experiments were conducted on the same samples using
commercial UMS HYPROP devices (UMS GmbH, Munich, Germany). The water retention and unsaturated hydraulic
conductivity data were then used to obtain parameters of the unimodal van Genuchten-Mualem model (van Genuchten,
1980; Mualem, 1976) by inverse modelling. For further details on parameters the reader is referred to appendix A.2.
The volumetric water content, $\theta$, was determined as the difference between sample weight and the oven-dry mass for
samples dried at 80°C until no difference in weight was measured (Gardner, 1986). Bulk density was determined as the ratio
of dry weight to the original sample volume. Volumetric water content at saturation, $\theta_s$, was assumed to be equivalent to the
sample total porosity. Saturated hydraulic conductivity ($K_s$; cm d$^{-1}$) was determined with a constant head test (Freeze and
Cherry, 1979) using the flow-through chambers described below and a hydraulic gradient of 1.



### 2.3 Saturated breakthrough experiments and inverse modelling

#### 2.3.1 Sample preparation

Saturated breakthrough experiments were conducted in 10 cm long, 10 cm i.d. Plexiglas™ (785 cm³) flow-through chambers, fitted at each end with 2.5x15x15 cm HDPE end-plates with silicon gaskets. A polypropylene fibre pad was placed between the plate and the sample to enhance the distribution of the solution beneath the sample (see appendix A.1). The NaCl solute source was a 20 l magnetically stirred solution reservoir pumped at a steady rate of 5 ml min$^{-1}$ (a flux density of 0.064 cm min$^{-1}$) using a peristaltic pump (WT600-3J, LongerPump, China) and the outflow solute concentration monitored continuously (e.g. Skaggs and Leij, 2002). Prior to the breakthrough experiment, the samples were flushed with 2 chamber volumes of the NaCl solution to reduce potential changes to the pore sizes as a result of swelling (Price and Woo, 1988; Ours et al., 1997) or clogging due to flocculation. Subsequently, the samples were inverted and flushed with ultra-pure water for 6 chamber volumes to remove the solutes that were introduced. To determine sampling times and the end of the experiment an EC electrode (11388-372, SympHony, VWR, USA) connected to a portable meter (SP80PC, SympHony, VWR, USA) was used The EC meter was calibrated using a 2-point calibration with 84 µS cm$^{-1}$ and 1413 µS cm$^{-1}$ conductivity calibration solutions (HI-7033 and HI-7031, respectively (Hanna instruments, USA). EC was checked every 5-10 minutes depending on the trend observed. Sampling was done with observed changes in EC and the experiment was continued until 1 hour after the outflow EC value was similar to the inflow. Samples were collected in 1.5 ml polypropylene micro centrifuge tubes (Z336769, Sigma Aldrich, USA) and kept frozen until analysed. Water samples were analysed for Na$^+$ and Cl$^-$ using an ion chromatograph (IC) (DIONEX ICS 3000, IonPac AS18 and CS16 analytical columns). Apparatus blank corrections were done as described in Rajendran et al. (2008), where no transport model for the apparatus blank was assumed but correction values generated using hermite cubic splines.

#### 2.3.2 Solute transport models

Two different parametric solute transport model types were used to describe the observed breakthrough data of Cl$^-$ and a third additional model for the Na$^+$ data. We list the model in the order of testing. The first two consisted of the mobile immobile equation (MIM; Eq. (1) and (2); van Genuchten and Wagenet, 1989) and the classical convection dispersion equation (CDE; Eq. (3); van Genuchten and Alves, 1982, Nielsen et al., 1986), and the third is the one-site adsorption equation (OSA; Eq. (5) and (6); van Genuchten et al., 1974, Nielsen et al., 1986) which was only used for Na$^+$.

The MIM for a non-reactive solute with instantaneous equilibration is given by

$$\beta\theta\frac{\partial c_m}{\partial t} = D\frac{\partial^2 c_m}{\partial^2 x} - v\frac{\partial c_m}{\partial x} - \alpha_{MIM}(c_m - c_{im}) \qquad \text{Eq. (1)}$$

$$(1-\beta)\theta\frac{\partial c_{im}}{\partial t} = \alpha_{MIM}(c_m - c_{im}) \qquad \text{Eq. (2)}$$





where $\beta$ is the ratio of the water content of the mobile region to the total water content, $\theta$ (L$^3$ L$^{-3}$), $C\_m$ and $C\_{im}$ are the
concentrations in the water phase of the mobile and immobile regions (M/L3), respectively, $D$ is the dispersion coefficient
(L$^2$ T$^{-1}$), $v$ is the average linear pore water velocity (L T$^{-1}$), and $\alpha\_MIM$ is the first order rate coefficient between the mobile
and immobile region (T$^{-1}$).
The CDE is given by

$$\frac{\partial c}{\partial t} = \frac{D}{R}\frac{\partial^2 c}{\partial^2 z} - \frac{v}{R}\frac{\partial c}{\partial z}$$
Eq. (3)


where $c$ is the concentration of the total water phase (M L$^{-3}$), and $R$ is a retardation factor for equilibrium adsorption, which
for a non-reactive solute is typically assumed to be 1 (but see Hoag and Price, 1997). In the classical interpretation, $R$ is
related to the adsorption distribution coefficient, $K\_d$ (M$^3$ L$^{-3}$), by R = 1+$\rho\_B$* K\_d/$\theta$. The MIM reduces to the CDE
equation under certain conditions, which can be analysed by the dimensionless Damkohler number ($D\_a$; Vanderborght et
al., 1997, Wehrer and Totsche, 1995), Eq. (4), given by

$$D_a = \frac{\alpha_{MIM} L}{v(1-\beta)\theta}$$
Eq. (4)


where $L$ is the column length (L). Large $D_a$ values indicate very fast equilibration between the regions. From inspection of
Eq. (4) it becomes clear that $\lim\limits_{\beta \to 1} D_a \to \infty$, and as $\alpha_{MIM}$ increases, $D_a$ increases proportionally, signifying instantaneous
equilibration, thus a differentiation between the two regions cannot be determined. Moreover, Parker and Valocchi (1986)
showed that the CDE may also be applicable when a considerable part of the solute dispersion is caused by diffusion into the
immobile region.
In physical non-equilibrium the attenuation of both reactive and non-reactive solutes is affected. However, if only the
reactive solute shows long tailing, then it can be assumed that chemical non-equilibrium is affecting the flow process. In
NaCl breakthrough experiments (Rezanezhad et al., 2012), only Na$^+$ showed long tailings in fen peat so that the physical
non-equilibrium model should not be employed. For this case, first-order kinetic chemical non-equilibrium models may be
chosen. One typical model for solute transport in porous media is the one-site adsorption equation which is an expansion of
the CDE with the addition of a kinetic adsorption member. It is given by

$$\frac{\partial c}{\partial t} = D\frac{\partial^2 c}{\partial^2 z} - v\frac{\partial c}{\partial z} - \alpha_{OSA}\left[(R-1)c - \frac{\rho_B}{\theta}s\right]$$
Eq. (5)

$$\frac{\rho_B}{\theta}\frac{\partial s}{\partial t} = \alpha_{OSA}\left[(R-1)c - \frac{\rho_B}{\theta}s\right]$$
Eq. (6)





where s is the kinetically sorbed concentration to the solid (M), and $\alpha_{OSA}$ is the first order rate coefficient between dissolved
and adsorbed concentration ($T^{-1}$) which has been found to be a function of pore water velocity and cannot be derived by
batch experiments alone (Nielsen et al., 1986).
Following the traditional approach for solute transport parameterization in peat (Rezanezhad et al., 2012, 2017), we initially
assumed the MIM model and compared it with the performance of the CDE for the non-reactive solute. For $Na^+$, transport
parameters were additionally estimated with a one-site chemical adsorption model. The data used for the fitting were
averages of three replicates. Parameterization of the model was done with CXTFIT (V2.0; Toride et al., 1995), which
minimises the least-squares. We estimated the following parameters: $v$ and $D$ for the CDE, $v$, $D$, $\beta$, and $\alpha\_MIM$ for MIM, and
$R$ and $\alpha\_OSA$ for the OSA model. The estimated $D$ and $v$ from the $Cl^-$ data fit were used for the $Na^+$ simulations, since
dispersion is a physical material property, and $R$ can only be determined if $v$ is fixed from knowledge of a conservative
solute experiment. Using various starting values, we ensured that the global minimum was found. CXTFIT calculates the
variance-covariance matrix, which is required for the calculation of the standard errors of the parameters and the parameter
correlation matrix.
The root-mean-squared-weighted error (RMSE) was used as an index for model performance calculated for each of the
tested models (Eq. (14) in Weber et al, 2016). The corrected Akaike Information Criterion (AICc; Eq. (2) in Ye et al. 2008)
was used as a method of model comparison where the model with the smallest AICc is to be favoured. It is a statistically
robust and commonly used index to compare models in soil physics (e.g. Weber et al., 2017a).

**2.4 Unsaturated breakthrough experiments and sensitivity analyses**

**2.4.1 Sample preparation**

The unsaturated solute breakthrough experiments were designed as six steady state evaporation columns 23 cm high and 10
cm i.d. (Fig.2). Peat samples were placed in a column constructed with a grooved HDPE base plate with an inlet, a silicon
washer and polypropylene fibre pad, and open at the top (see appendix A.1). The columns were slowly saturated from the
bottom in small increments over 48 hours to minimize trapped gas bubbles. Once saturated, the columns were flushed with 2
column volumes of the NaCl solution to reduce potential changes in hydraulic properties, as previously described. This was
followed by flushing 6 column volumes of ultra-pure water to remove the $Na^+$ and $Cl^-$, with the water overflowing from the
top of the sample. Flushing of $Na^+$ did not remove all solute, resulting in 40 mg $l^{-1}$ of $Na^+$ remaining in the time 0 samples
taken at the bottom 8 cm of the column. Nevertheless, these concentrations were accounted for in the HYDRUS simulation.
Columns were then drained overnight with 0 cm pressure head at the bottom of the sample, and connected to a Marriot
device containing ultra-pure water. The water table was set to the base of the peat column, and the columns were left to settle
for 11 days, after which the columns were instrumented with the soil tensiometers and water samplers (further details
below).



Each Marriot device was fitted with a low flow 12-volt mini water pump to circulate the water within it for 5 minutes every
two hours, to prevent solute stratification. Three Marriot devices were filled with an 8.9 mM solution of NaCl as treatment
and three with ultra-pure water. The columns were fitted with 4 unsaturated soil water samplers at 2.5, 7.5, 12.5 and 17.5 cm
above the water table (19.21.05, Rhizon, Rhizonsphere, Germany), and with 2 two tensiometers at 10 cm (LM) 23 cm (UM),
to determine if the water pressure deviated from hydrostatic conditions (see appendix A.3). Tensiometers were composed of
a porous clay cup and a flexible silicon tube, which was open to the atmosphere. The experiment was run for 120 days;
evaporation was calculated based on changes to the water level in the Marriot device (see appendix A.3). The experiment
was conducted in a room with controlled humidity maintained at ~45%, assisted by a fan to mix the air in the chamber.
Relative humidity (RH) and temperature were measured every 10 minutes (ECT, Decagon, USA) (see appendix A.3).

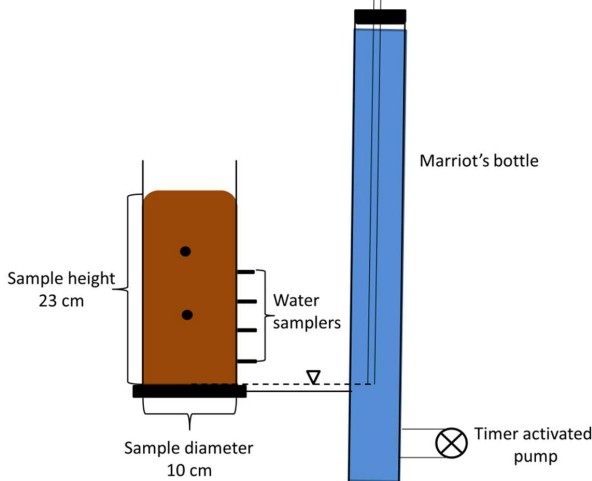

● Soil water pressure meter


**Figure 2 – Unsaturated column experiment column and water reserve setup.**

NaCl solution was introduced at the base of the column and drawn upwards by evaporation. Marriot devices were attached to
supply water over the bottom boundary for each column. The pressure head at the lower boundary was fixed to a pressure
head of 0 cm for the duration of the experiment. Daily measurements of the water level in the Marriot were measured with a
measuring tape, and the evaporative water flux over the upper boundary was calculated by dividing the water lost by the
surface area of the column (see appendix A.3).





Pore-water samples were taken weekly from each sample height through the Rhizon samplers. On average, 5.5 ml of water
was drawn from each sampler using a dedicated 30 ml polypropylene syringe (Z683647, Sigma-Aldrich, USA). To ensure
equal pull on each sampling point, 6x4x4 cm HDPE spacers were fabricated and placed within the syringe body and piston.
Only samples from time steps 0, 21, 42, 63, 84 and 120 days were analysed. After the experiment ended the cores were
frozen then sliced to ~2 cm thick sections using a band saw. Sections were measured with callipers, weighed and placed in
pre-weighed, food grade and heat resistant bags. The slices were than thawed and ultra-pure water, twice the weight of the
slice, was added to extract the solutes and placed on a table shaker (MaxQ 3000, Thermo scientific) for 48 hours. All
samples were frozen until analysed for $Na^+$ and $Cl^-$ via IC at the Biogeochemistry Laboratory at the University of Waterloo
(DIONEX ICS 3000, IonPac AS18 and CS16 analytical columns). Results were adjusted to account for the dilution effect of
the added water.

### 2.4.2 Numerical simulations and sensitivity analyses

The steady state unsaturated evaporation experiment with solute transport was simulated with HYDRUS-1D (Simunek et al.,
2008), which numerically solves the Richard's equation for water flow, and the solute transport equations. For the water
flow, the soil hydraulic properties are the necessary input and were parameterized with a unimodal van Genuchten-Mualem
equation using data collected in the tension disk experiments and transient evaporation experiments. The model domain
represented the 23-cm high column with a spatial discretization of 0.5 cm.
The lower boundary condition for the water flow was a constant zero pressure representing the water table. The upper
boundary condition was a flux boundary based on measured evaporation rates. For the solute transport, the lower boundary
was a fixed concentration in the liquid phase and the upper boundary condition was a zero flux. To account for the soil
solution sampling (that would otherwise lead to a misrepresentation of water flow and solute transport), we used the Root
Water Uptake function in HYDRUS by specifying an individual root at the height of each of the four Rhizon samplers. The
applied root water uptake model (Feddes et al., 1978) assumes no salinity stress, no pressure dependent reduction of given
water uptake quantity, and a quasi-infinite maximum allowed concentration for passive root solute uptake. The total water
volume extracted per sampling day was taken to be equal at each root node.
Dispersion is dependent on the average linear velocity, which in turn is dependent on the water content (Perkins and
Johnston, 1963). To date, HYDRUS does not account for a water content dependency of dispersion ($D(\theta)$) in solute
transport modelling. Also, this information is not available from the saturated experiment. Therefore, a sensitivity analysis of
D on the model response was done to approximate its quantitative influence on the solute transport. To gauge the range of
values for the sensitivity analysis, a calculation of the change in D was performed using data gathered from the unsaturated
columns. The calculation used the equation for D in capillary flow under unsaturated conditions in soils as a function of v by
Fried and Combarnous (1971) (not shown). The equation connects D to changes in water content and allows the calculation
of changes in D due to measured changes in water content. Comparison of the calculated values provided with a range of





change in D. Calculations indicated that the change in D ranged from 8% to 15%. Therefore, to add extra range the
sensitivity analysis for the HYDRUS model was performed using ±20% and a ±100% change in D.

## 3 Results and discussion

### 3.1 Soil physical properties

The bulk density and porosity of the peat used in the various experiments was similar (Table 1). The retention curve (Fig. 3)
does not have the classical shape that would point at discrimination between an active and inactive porosity (Rezanezhad et
al., 2016, Weber et al., 2017b). Measured water retention and hydraulic conductivity data closely fit the van Genuchten-
Mualem unimodal model with an RMSE of 0.03 and AICc value of -342 (Fig. 3; Table 2). Furthermore, the estimated Ks
value of the unimodal fit (106 cm d$^{-1}$: Table 2) is similar to the measured value (100 cm d$^{-1}$; Table 1).
**Table 1 - Soil physical and hydraulic properties of prepared peat cores from different experiments. Values are averages;**
**percentages in brackets are the coefficient of variation. Porosity was calculated using particle density from Ketcheson et al. (2017).**

| Type of experiment | n | $\rho_B$ | $K_s$ | Porosity |
|---|---|---|---|---|
| | | [g cm$^{-3}$] | [cm d$^{-1}$] | [-] |
| **Saturated breakthrough** | 3 | 0.12 (1.7%) | 99.7 (2.1%) | 0.93 (1.3%) |
| **Unsaturated columns** | 6 | 0.12 (4.1%) | | 0.93 (2.8%) |
| **Retention** | 4 | 0.12 (3.3%) | | 0.93 (2.3%) |




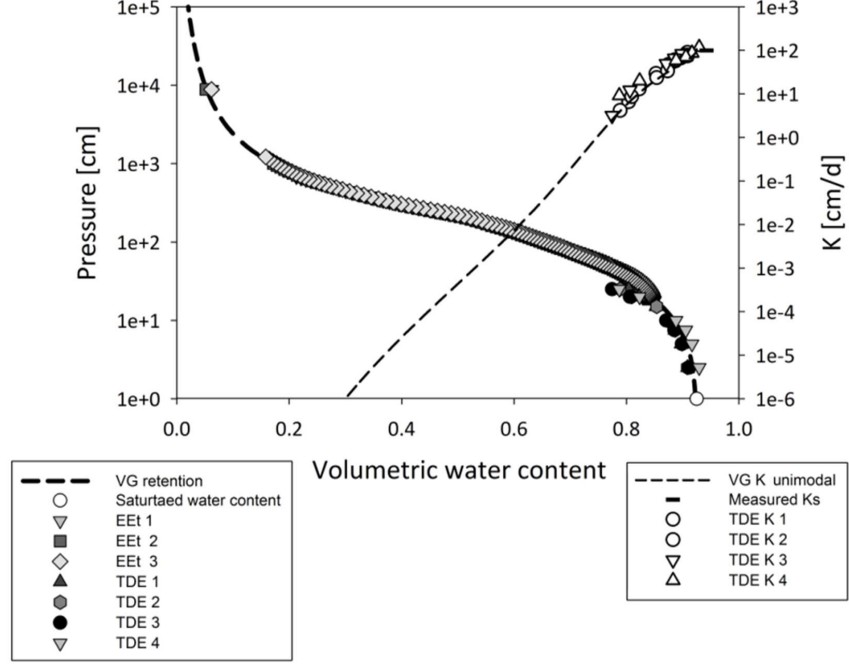


**Figure 3 – Soil water retention and hydraulic conductivity curves with measurement results of the transient evaporation experiments (EEt) and Tension Disk Experiments (TDE) and parameterizations for the unimodal van Genuchten-Mualem model. Negative pressure was used for the retention experiment.**

**Table 2– Parameter results for the soil hydraulic properties function of the unimodal model saturation function; the parameter**
**names are explained in the text.**

| model | $\theta_r$ | $\theta_s$ | $a_1$ | $n_1$ | $K_s$ | $\tau$ | $w_2$ | $a_2$ | $n_2$ | $n_p$ | RMSE$\theta$ (h) | RMSE $\log_{10}$ K(h) | AICc |
|---|---|---|---|---|---|---|---|---|---|---|---|---|---|
| | [-] | [-] | [cm$^{-1}$] | [-] | [cm d$^{-1}$] | [-] | [-] | [cm$^{-1}$] | [-] | [-] | [-] | [cm d$^{-1}$] | [-] |
| uni | 0 | 0.93 | 0.015 | 1.6 | 106 | 10 | - | - | - | 5 | 0.03 | 0.19 | -342 |


## 3.2 Saturated breakthrough experiment

Cl⁻ breakthrough started around ~60 minutes with C/C$_0$=0.5 arriving 97 minutes from the start of the experiment
(Fig. 4). Complete Cl⁻ breakthrough (C/C$_0$=1) was achieved after 300 min. Similar to Cl⁻, initial Na⁺ breakthrough began ~60
minutes from the start of the experiment (Fig. 4). However, C/C$_0$=0.5 was not achieved until ~250 minutes, with only ~0.85
breakthrough at the end of the experiment that had prolonged tailing, indicating non-equilibrium process. The EC curve is
similar in shape to that of Cl⁻, but took longer to reach the full breakthrough. Attenuation of $Na^+$ compared to Cl⁻ is evident
by the greater time until $C/C_0=0.5$ (Fig. 4), is attributed mainly to the high adsorption capacity of peat (Ho and McKay,
2000). In contrast, Cl⁻ attenuation in peat is mainly due to mechanical dispersion and diffusion into dead-end pores and not
adsorption (Price and Woo, 1988). The dissimilarity of the EC breakthrough curve to that of $Na^+$ (Fig. 4) demonstrates the
limitation of using EC as an indicator for reactive solutes. This limitation is due to enrichment of ions in the solution from
the soil and cation exchange with the medium, which changes the solution concentration of the cation of interest; therefore,
EC can be a good estimator for non-reactive solutes but is limited as an indicator for cation transport (Olsen et al., 2000;
Vogeler et al., 2000).

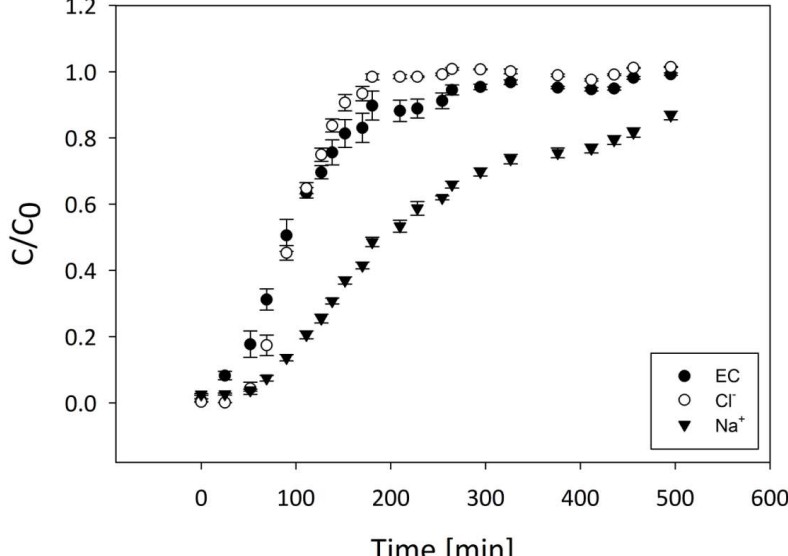


**Figure 4 – EC, Cl⁻ and $Na^+$ corrected saturated breakthrough curves in saturated peat over time. Each point is an average of 3**
**samples, error bars are standard error of mean. Errors were not accounted for in the fitting.**

### 3.3 Unsaturated column experiment

The evaporation rate of the experiment was 3 mm d⁻¹ (not shown, see appendix A.3). As expected, Cl⁻ was transported faster
than $Na^+$ as evident by the more rapid rise of Cl⁻ in the peat profile (Fig. 5). $C/C_0=0.5$ of Cl⁻ reached 7.5 cm above water



table within 21 days (Fig. 5a) and by 42 days $C/C_0$=0.5 reached 17.5cm (Fig. 5a). Complete breakthrough ($C/C_0$=1) of $Cl^-$
was achieved between 63 to 84 days from start of experiment (Fig. 5a).

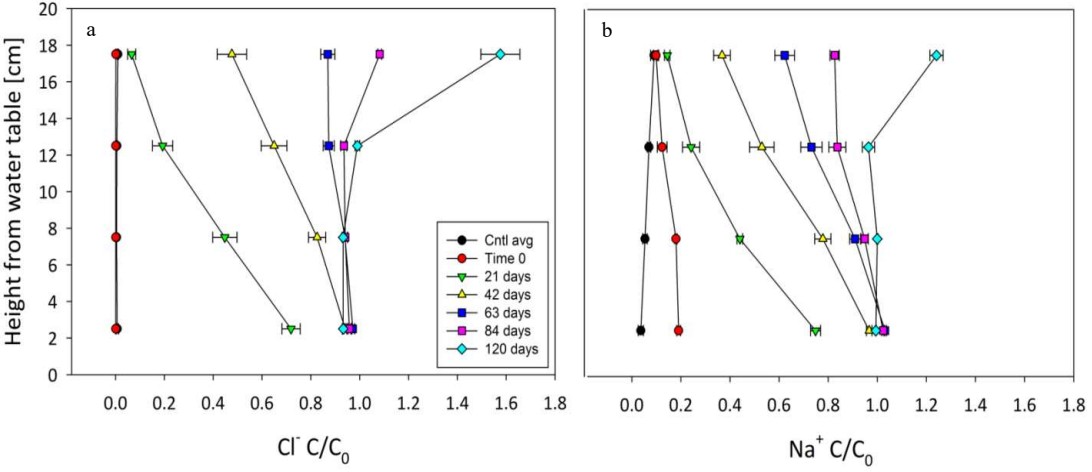


**Figure 5 - Breakthrough curves of solutes in the unsaturated columns profile, a) of $Cl^-$, and b) of $Na^+$. Values presented are**
**averages and whiskers are standard errors. "Cntl avg" represents the average of control measurements; for this aim, all**
**measurements in a specific height were averaged with each point representing 18 measurements. For the treatment, each point is**
**an average of 3 measurements. Each treatment curve represents a different sampling time from start of experiment. 0 cm is the**
**water table location.**
Comparably, $Na^+$ $C/C_0$=0.5 reached 7.5 cm within 21 days (Fig. 5b). After 42 days, the $C/C_0$=0.5 was located between 12.5
to 17.5 cm from the water table (Fig. 5b). Complete $Na^+$ breakthrough occurred later than $Cl^-$, sometime after 84 days but
before 120 days (Fig. 5b). The accumulation of both elements above inflow concentrations (C/C0>1) at 17.5cm after 120
days (Fig. 5b), indicates evaporative accumulation occurred as water molecules left the column while solute molecules
remained (Tsypkin, 2003). Therefore, evaporative accumulation enhances the breakthrough rate as ions remain in the soil
while water evaporates; thus, producing a faster accumulation rate than if the breakthrough was estimated using a saturated
flow system where the solutes would leave the system with the solution. Nevertheless, this effect is a basic product of
evaporation controlled transport (Elrick et al., 1994; Tsypkin, 2003).

### 3.4 Simulations

#### 3.4.1 Solute transport model selection

For all transport models, the fitted parameters and associated uncertainties, AICc, and RMSE values are given in Table 3.
The CDE and MIM model for $Cl^-$ fit the data well (Fig. 6a), and have identical RMSE (0.032 mg/l). However, the AICc is



favors the CDE. Additionally, the MIM model estimated parameters (v, D and β) show much larger coefficients of variation,
with β varying by 1510% (Table 3). During fitting, $\alpha_{MIM}$ ran into the CXTFIT internal upper boundary, further suggesting
that the application of the MIM is an over-parameterization and therefore not suitable. Also, the MIM has two additional
parameters than the CDE model, which makes the simpler CDE model preferable (Cavanaugh 1997).
The Peclet number for the fitted Cl⁻ breakthrough data, which is the ratio of advective vs diffusive transfer, was 33.9. In
systems with values > 2 diffusion is considered negligible (Huysmans and Dassargues, 2005). Moreover, with $\beta \rightarrow 1$ (Table
3), the Damköhler number, $D_a$ approaches infinity, so that the equilibration between the mobile and immobile zones is
considered instantaneous (Wehrer and Totsche, 2005; Vanderborght et al., 1997). In other words, $D_a$ indicates the system is
not governed by physical non equilibrium processes, but rather the simpler CDE concept applies. The significance of this is
that the physical non-equilibrium approach may be excluded for these samples and boundary conditions. One reason for this
finding could be based on the inherent nature of the samples. The peat of the Nikanotee Fen watershed was moderately
decomposed sedge peat containing small amounts of *Sphagnum* moss (Nwaishi et al., 2015). It is the *Sphagnum* mosses that
contain the hyaline cells (Hayward and Clymo, 1982), which are probably the main cause for the existence of dead end
pores. Therefore, with only a small part of the peat originating from *Sphagnum*, the potential for dead end pores was small
compared to peat that originates mainly from *Sphagnum* moss. Additionally, evidence found in the SEM scans of the peat
used in this study (Fig. 1), shows that the cell walls have decayed, with only the skeleton of the cell remaining, while the
skeleton itself is still intact so water can move much more freely through these structures. These results contradict the
hitherto assumption that solute transport in peat has to be simulated using the MIM. Additionally, this is reflected in the fact
that a multimodal retention curve was not observable, which would have been indicative for a two domain flow of solute
transport.
As the CDE model provides a good description of the saturated Cl⁻ breakthrough, and physical non-equilibrium can be
discarded as the underlying process (Table 3; Fig. 6b). Therefore, the non-equilibrium effect observed in the Na⁺
breakthrough (Fig. 6b), must be due to chemical processes such as an interaction of Na⁺ ions with negatively charged sites
on the peat surface whether through retardation or adsorption. Having shown that the MIM is not parsimonious in its
parameters and the robust estimates of *v* and *D* for the CDE, these were fixed when fitting the remaining model parameters
of the CDE and one-site adsorption model for Na⁺. First, the CDE was fitted with *R* to the Na⁺ data; the resulting curve
shows that equilibrium adsorption does not fit (Fig. 6b). In comparison, the one-site adsorption model fit well (Fig. 6b) and
had a lower RMSE and a considerably lower AICc value (Table 3). Based on the estimated R-value of the one-site
adsorption model, the $K_d$ value of Na⁺ was 15.6 l kg⁻¹. The parameters from the CDE for Cl⁻ and from one-site adsorption for
Na⁺ were then used for the HYDRUS simulation of the unsaturated columns.



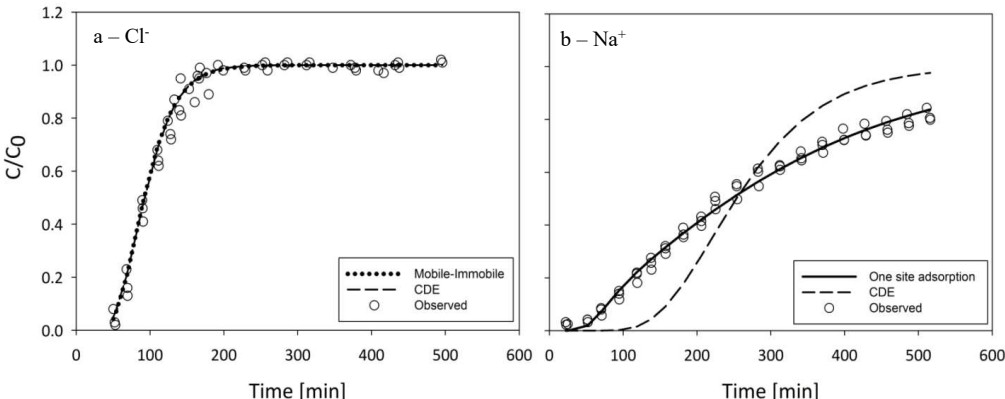


**Figure 6 –Breakthrough curves of observed values and fitted models. a) of Cl⁻ ; b) of Na⁺.**


**Table 3 – Estimated saturated transport parameters and the models' goodness of fit data. N.A. = not applicable. Estimated values are presented with coefficient of variation as percentages in brackets, the parameters are explained in the text.**

| Solute | Model | v | D | R | β | α | RMSE | AICc |
|--------|-------|---|---|---|---|---|------|------|
| | | (cm min⁻¹) | (cm² min⁻¹) | - | - | (min⁻¹) | (mg l⁻¹) | (-) |
| Cl⁻ | MIM | $9.81*10^{-2}$ (91%) | $6.66*10^{-2}$ (19%) | fixed to 1 | 1.00 (1510%) | $6.05*10^{-1}$ (0%) | 0.032 | -406 |
| | CDE | $9.79*10^{-2}$ (1%) | $6.66*10^{-2}$ (7%) | fixed to 1 | N.A. | N.A. | 0.032 | -408 |
| Na⁺ | CDE | fixed | fixed | 2.65 (3%) | N.A. | N.A. | 0.145 | -229 |
| | OSA | fixed | fixed | 3.07 (1%) | N.A. | $6.71*10^{-4}$ (3%) | 0.024 | -443 |


### 3.4.2 Unsaturated column simulations and sensitivity analyses

The HYDRUS predictions of solute concentrations at the four observation points were good for both solutes (Fig. 7), even though the solute transport model parameterization was based on the saturated experiments. Plotting of the concentrations from the solute extractions for the upper part of the core at the end of the experiment reaffirmed the models' generally good fit for both solutes (Fig. 7), although in both cases the models underestimate the measured concentration at the very top of the soil profile (Table 4).





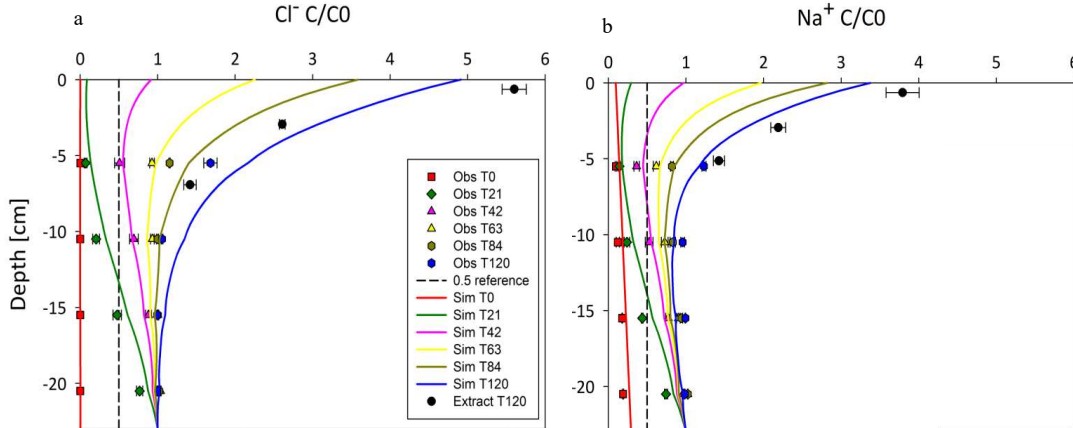


**Figure 7 - Observed values (Obs) from the unsaturated column experiment vs simulated values (Sim) of a) Cl⁻; and b) Na⁺. Observed values are averages and standard error, n=3. T stands for time and the number that follows is the number of days. Extract T120 represents values measured via extraction as part of post experiment processing. Zero (0)0 depth marks the surface of the column. Dashed reference line marks C/C0=0.5.**



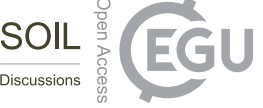

**Table 4 - Unsaturated transport parameters used in or estimated by HYDRUS and models' goodness of fit data. -- = not**
**applicable. Estimated values are presented with the coefficient of variation as percentages in brackets. Diff. W is the molecular**
**diffusion coefficient of the solutes.**

|  | D [cm² min⁻¹] | Kd [l kg⁻¹] | *Diff.W [cm² min⁻¹] | α [1 min⁻¹] | β | RMSE [mg l⁻¹] |
|---|---|---|---|---|---|---|
| **Cl⁻** | $6.81*10^{-2}$ | 0 | $1.22*10^{-4}$ | -- | -- | 15.65 |
| **Na⁺** | $6.81*10^{-2}$ | 15.6 | $7.98*10^{-5}$ | $1.11*10^{-2}$ (1.1%) | 1.00 (8.1%) | 10.19 |

* taken from Appelo & Postma (2004).


Given that the dispersion coefficient for the unsaturated modeling was based on measurements in the saturated flow-through
chambers, a sensitivity analysis was performed with HYDRUS to determine its impact on the simulations. It indicates that a
±20% change in the dispersion coefficient resulted in a ±1.2% and a ±4.1% change in the final concentration of Cl⁻ and Na⁺,
respectively (Fig. 8). Further, an analysis with a ±100% change in D altered the final concentrations by -5% to 6.5% for Cl⁻
and by 9% to -17% for Na⁺. The analysis demonstrates unsaturated transport is not highly sensitive to changes in the
dispersion coefficient under the experimental conditions used. Furthermore, since the differences in water contents were not
large, ranging between 0.93 at full saturation to 0.84 at the top of the column, it is likely that the actual hydrodynamic
dispersion did not vary significantly.

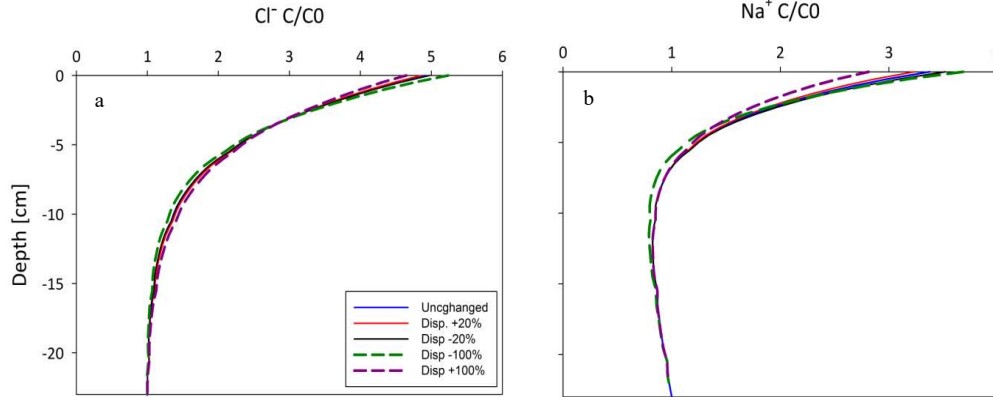


**Figure 8 – Sensitivity analysis of unsaturated transport for changes in the dispersion coefficient. a) in Cl⁻ transport, b) in Na⁺.**





**Conclusions**
Saturated breakthrough experiments on disturbed peat, sampled from the Nikanotee Fen, were conducted in the laboratory
using conservative and reactive solutes. With this we tested if the common assumption of mobile-immobile solute transport
process best reflects the transport processes in saturated and unsaturated peat. Based on inverse modelling of time series of
measured conservative tracer concentrations, and robust statistical evaluation, we found that the MIM model was an over-
parameterization for $Cl^-$, since very good results were found using the simpler CDE.
For this reason, it could be deduced that the $Na^+$ attenuation, expressed by prolonged tailings in the observed breakthrough in
the fen peat is chemically based, as the physical non-equilibrium (i.e. MIM) approach would have had an effect on both
solutes. Hence, we can conclude that $Na^+$ showed distinct chemical non-equilibrium adsorption process, which could be
described using the OSA model, and still fulfilling the requirement of parsimony. The results are in contrast to the
commonly accepted MIM behavior of solutes breakthrough in peat samples.
The significance of this is that while reactive solutes may be heavily attenuated in peat, conservative solutes are not
necessarily retarded (Hoag and Price, 1997). In this case the degraded structure of the peat (Fig. 1) eliminated many of the
enclosed spaces commonly visible in less decomposed *Sphagnum* peat (see Hoag and Price, 1997; Rezanezhad et al., 2016).
Measured water retention data was adequately described using a unimodal expression for the underlying pore size
distribution, corroborating the finding that a physical dual porosity structure was not present.
On a side note, we can attest that the use of EC as an indirect measurement for a reactive solute will result in overestimation
of breakthrough if the solute interacts with the solid phase.
This research implies that automatically assuming mobile and immobile regions in peat is incorrect. The sedge peat with
remnants of *Sphagnum* moss used in this experiment potentially had a limited amount of dead end pores, due to the low
content of Sphagnum moss with hyaline cells. Furthermore, evidence suggests that the peat used has decayed enough to lose
the cell walls but not enough to break the cell skeleton, and is likely why the peat lacks the classically assumed MIM
regions. The decomposition may have been enhanced by aeration of the peat in the donor fen (Nwaishi et al., 2015).
Additionally, it is concluded that transport parameters gathered in saturated breakthrough experiments can be used to
simulate transport in slightly unsaturated media under near steady state conditions. Data gathered show that the accumulation
of solutes via evaporation causes concentration to rise quickly above the initial concentration. While these results are valid
for the described boundary conditions and initial conditions, the fate of salt accumulation is not clear under more natural
conditions such as complex meteorological evapotranspiration-precipitation cycles, with, for example, surface inundation
and overland flow export of solutes. Additionally, different salt concentration levels at the lower boundary of the experiment
were not investigated, which has been documented in the case of the Nikanotee Fen watershed (Kessel, 2016). As a first
assessment of the effect $D$ has on salt accumulation, a synthetic parameter sensitivity analyses was carried out for $Na^+$. To
further understand the rates of the evaporative accumulation, more complex numerical transport model should be used,
including flushing due to precipitation and runoff, using the parameters reported in this study along with various weather



scenarios. Considering the complex hydraulic retention and conductivity properties of *Sphagnum* mosses and peat, it is
conceivable that a wide range of tested water contents could affect the choice of the underlying transport process.
Additionally, the experiment was carried out under steady state conditions, unlike the complex meteorological patterns in the
field. Finally, the implications for reclamation projects are that if one of the goals is to enhance solutes attenuation, the origin
and composition of the peat, its water retention properties along with its decomposition state should be characterized as not
all peats will perform equally. From the industry perspective, choosing and peat with dead end pores would allow a potential
for significant attenuation.

**A.1 Appendix 1: Pictures of saturated and unsaturated experiments.**

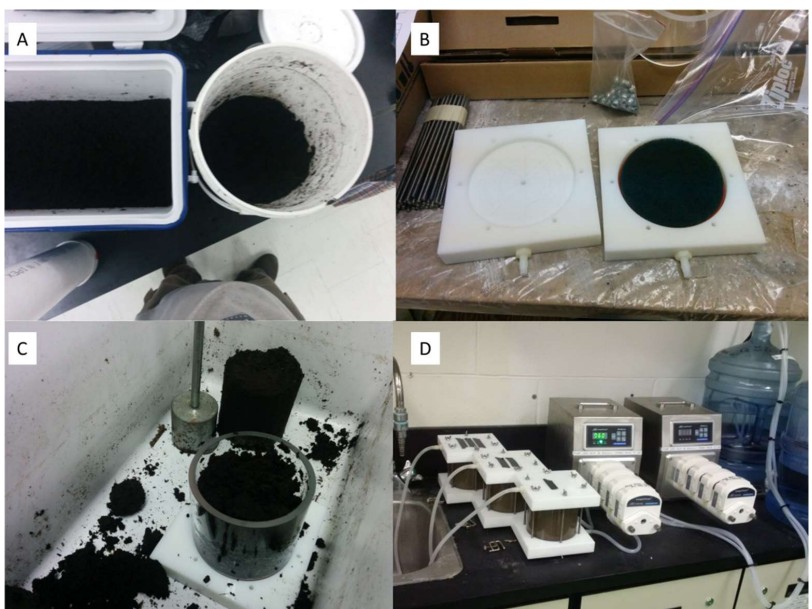


**Figure A.1.1 - Pictures from saturated transport experiment. A) Cleaning and mixing the peat; B) flow through cells plates. The**
**green pad is below the sample, redistributing the water beneath it; C) packing cell with peat; and D) flow through experiment**
**setup, cells are connected to a pump drawing the solution from a container on a magnetic stirrer.**




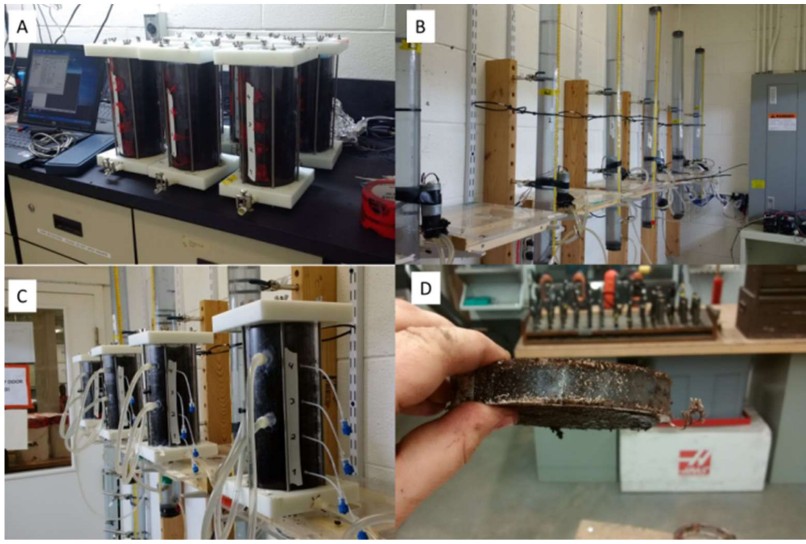


**Figure A.1.2 - Pictures from the unsaturated transport experiment. A) columns with peat, note the laptop for scale; B) Marriot**
**bottles and pumps; C) columns connected and instrumented, Blue caps are the soil pore water samplers, large tubes are the**
**tensiometers; and D) slice of a peat column before extraction at end of experiment.**

**A.2 Appendix 2: Soil hydraulic properties**
Measurements of water retention properties
The tension disk experiment (TDE) was conducted on 10 cm i.d. and 5 cm high peat samples at seven different pressure
steps under unsaturated unit gradient vertical flow conditions using a tension disk apparatus that used 15 μm Nytex screens
to prevent air entry below the air entry pressure (~35 cm) (Price et al., 2008, McCarter et al., 2017). Samples were initially
saturated for 48h and two layers of cheese-cloth covered the top and bottom of the sample to maintain the integrity of the
surfaces. The pressure steps ($h$; cm) were -2.5, -5, -7.5, -10, -15, -20, -25 cm, which was also the order in which the
experiment was conducted. During the experiment, outflow was monitored for each pressure step by a scale with an accuracy
of at least 0.1 g and logged at 1-minute intervals. The experiment stopped when there was no change from past
measurements over a 30-min. period. After each step the weight of the sample was determined to enable calculation of the
water content. From the outflow, the unsaturated hydraulic conductivity was calculated from the Darcy-Buckingham
equation (Swartzendruber, 1969 ).
**Saturated hydraulic conductivity**



The FTC were used for the determination of the saturated conductivity ($K_s$; cm d⁻¹), too, which were connected to a
Marriot's bottle supplying a constant pressure head. The adopted method was a constant head test (Freeze and Cherry, 1979)
with a gradient of 0.44. Once the outflow stabilized, it was measured in a 250ml glass graduated cylinder (S63459, Fischer
Scientific, USA) every 2 min over 20 min.
**Transient evaporation experiment**
The transient evaporation experiment (EEt) was conducted on the same samples as the TDE. With a 0 cm pressure head at
the bottom prior to the beginning of the EEt with the commercial UMS HYPROP device (UMS GmbH, Munich, Germany).
The samples had a larger diameter than the UMS HYPROP device so that Plexiglas screens were used at the bottom to seal
and prop the sample. The pressure head was directly measured in the middle of the sample and, thus, directly related to the
calculated water content to obtain the retention information, which is a valid approximation at or near a linear pressure
distribution (Becher, 1971). With this, the evaluation for conductivity is not reliable.
**Inverse fitting of soil hydraulic properties**
The water retention and unsaturated hydraulic conductivity data were used to parameterize soil hydraulic property (SHP)
models. We used the unimodal van Genuchten-Mualem model combination (van Genuchten, 1980; Mualem, 1976). We used
the analytical expression derived by Priesack and Durner (2006). The soil water retention function is given by

$$\theta(h) = \theta_r + (\theta_s - \theta_r)\,\Gamma(h) \qquad\qquad \text{Eq.(A.1)}$$

where $\theta_r$ is the residual and $\theta_s$ the saturated water content (cm³ cm⁻³) and $\Gamma(h)$ (-) the effective saturation given by

$$\Gamma(h) = \sum_{i=1}^{k}\Gamma_i(h) = \sum_{i=1}^{k} w_i\,[1 + (-\alpha_i h)^{n_i}]^{m_i} \qquad\qquad \text{Eq.(A.2)}$$

where, $w_i$ is a weighting coefficient between the modal pore size distributions, and $\alpha_i$ (cm⁻¹) and $n_i$ (-) are shape parameters
with constraining, $m_i = 1 - 1/n_i$. The unimodal van Genuchten saturation function is obtained by $k = 1$. The unsaturated
hydraulic conductivity is expressed as

$$K(\Gamma) = K_s \Gamma^{\tau} \left(\sum_{i=1}^{k} w_i \alpha_i\right)^{-2} \left(\sum_{i=1}^{k} w_i \alpha_i \left[1 - \left(1 - (\Gamma)^{1/m_i}\right)^{m_i}\right]\right)^{2} \qquad\qquad \text{Eq.(A.3)}$$

where $K_s$ is the saturated hydraulic conductivity (cm d⁻¹) and $\tau$ is sometimes referred to as a tortuosity constant, which
should be positive-
All parameters were estimated except for $\theta_s$, which was set to 0.925, i.e. the porosity value. Estimation was done in R.3.2.1
(R Core Team 2015) with implementation of the differential evolution optimiser to minimise the sum of squared errors for
the retention and hydraulic conductivity curves (Mullen et al., 2011). The estimation of the soil hydraulic properties of the
fen peat by inverse estimation was done as described in Peters and Durner (2008). After all procedures were concluded, bulk
densities for all samples were determined gravimetrically based on an oven-dry mass basis for samples dried at 80 °C until
no difference in weight was measured (Gardner, 1986). From knowledge of the dry weight and experimental system weight
water contents could be calculated for the soil hydraulic properties.



Statistical parameters
The root mean square error (RMSE) is used as a metric to describing the model prediction quality, such that

$$RMSE = \sqrt{\frac{1}{m}\sum_{l=1}^{m}(y_l - \hat{y}_l)^2}$$    Eq.(A.4)


where $m$ is the number of observations, $y_l$ is the observed and $\hat{y}_l$ the model predicted value (solute concentration, water
content or hydraulic conductivity). The corrected Akaike Information Criterion (AICc; Ye et al. 2008)) was also used as a
method of model comparison where the model with the smallest AICc is to be favored. Applications so soil hydrological
model testing can be found in Weber et al. (2017a, 2017b).

$$AICc = m\,\ln\left(\frac{1}{m}\sum_{l=1}^{m}(y_l - \hat{y}_l)^2\right) + 2\,n_p + 2\,\frac{n_p(n_p + 1)}{m - n_p - 1} + C$$    Eq.(A.5)

where $n_p$ is the number of parameters of a respective model.
Estimation of model parameters was done by minimising the sum of squared errors for the retention and hydraulic
conductivity curves in R.3.2.1 (R Core Team 2015) with an implementation of the differential evolution optimiser (Mullen et
al., 2011) by adopting the multi-objective function as described in Peters and Durner (2008).

References (References that do not appear here can be found in the main reference list):
Price, J. S., Whittington, P. N., Elrick, D. E., Strack, M., Brunet, N., & Faux, E. (2008). A method to determine unsaturated
hydraulic conductivity in living and undecomposed moss.  Soil Sci. Soc. Am. J. , 72(2), 487-491.


**A.3 Appendix 3: Unsaturated experiment conditions**

510        Ambient conditions in the chamber where the experiment was conducted were stable with an average temperature

of ~25±1 $^0$C and an average RH of 41±0.02 % (Fig. A.3.1). Additionally, water pressure profile in the soil did not vary much
for each column; soil water pressure above the water table averaged -10.7±0.5 cm for the low meter and -19.5±0.5 cm for the
high meter (Fig. A.3.2). Furthermore, data from all columns were in a similar range (Fig. A.3.2) meaning the columns were
reasonable replicates in soil water pressure.



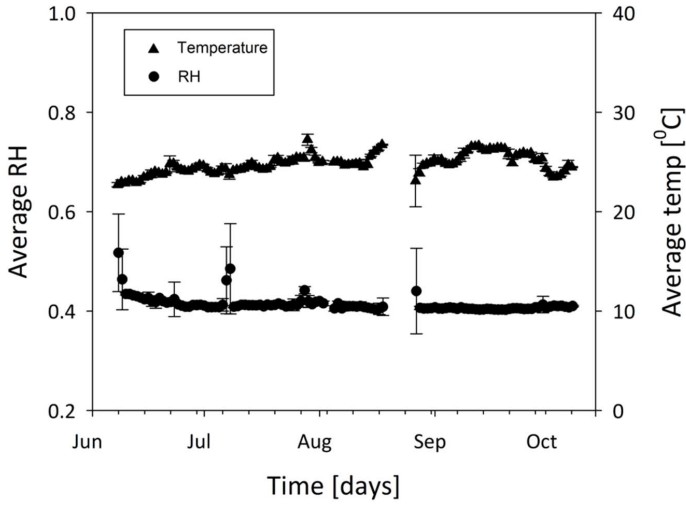


**Figure A.3.1 – measured temperature and relative humidity during the unsaturated column experiment. Each point is a daily**
**average of 144 measurements and corresponding standard error.**

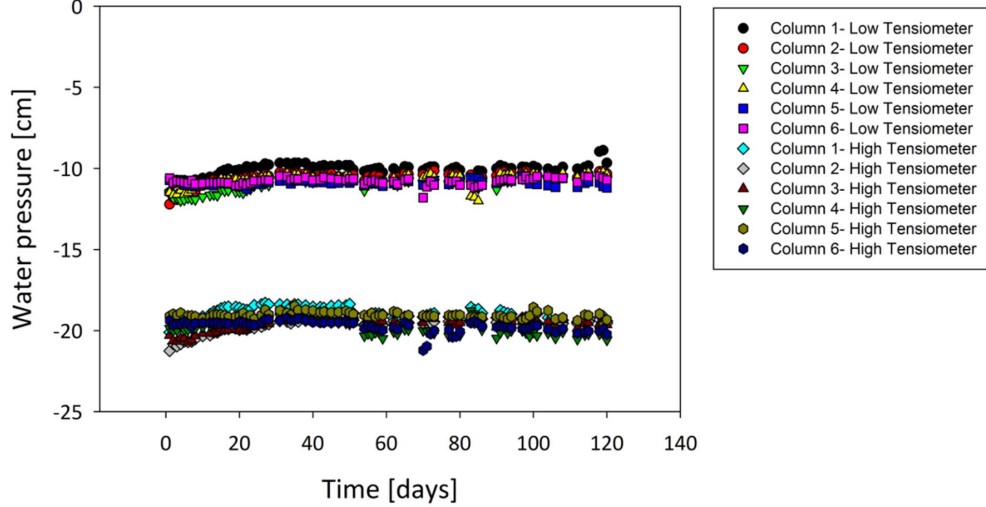




**Figure A.3.2 – Soil water pressure measurements over time. 0 cm marks the water table. Values around the -10 cm mark are from**
**the low pressure meters of all 6 columns; values around -20cm are from the high pressure meters.**

521         Moreover, E data strengthen the conclusion that the columns were decent replicates with an overall low fluctuation

in values averaging at $0.27\pm0.05$ cm d$^{-1}$ (Fig. A.3.3).

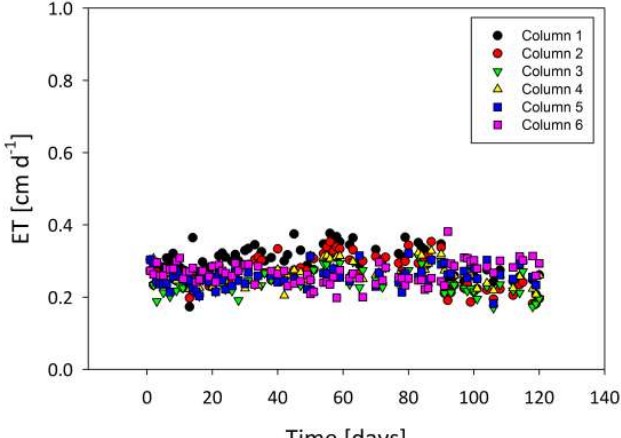


**Figure A.3.3– Calculated ET during the experiment. Data presented is for each column separately (color coded, see legend).**


**Acknowledgements**
Thank you to Harmen Vander Heide, Dan Beaver, Andrew Urschel, Scott Ketcheson, Eric Kessel, Tasha-Leigh Gauthier,
James Sherwood, Corey Wells and Vito Lam for their help in logistic, fabrication and analysis. Additionally, thanks to Dr.
Mazda Kompanizare, Behrad Gharedaghloo and Dr. Fereidoun Rezanezhad, and Dr. SC Iden for their advice. Funding from
the following sources is gratefully acknowledged: Natural Science and Engineering Research Council Collaborative
Research and Development (NSERC-CRD), Suncor Energy Inc., Shell Canada Limited and Esso Imperial Oil Limited, and
the German Academic Exchange Service.

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
