# Peer review of "Saturated and unsaturated salt transport in peat from a constructed"

_SOIL, 2017_

## Referee Comment (RC1) · Anonymous Referee #1 · 10 Aug 2017

Hydraulic as well as solute transport properties of peat soils are still poorly understood as compared to mineral soils. In their study the authors address both hydraulic properties and solute transport parameters. Unfortunately, none of the mechanisms is considered with the necessary depth to reveal new insight. The paper suffers from the treatment of too many different processes and employed methods. Moreover, an unjustified experimental approach and shortcomings in the model evaluation do not allow drawing definite conclusions.

The paper shall not be published because of a confusing number of considered aspects. A re-consideration of the manuscript makes sense only if additional experiments are conducted and/or emphasize is laid on either hydraulic properties or solute transport conditions and if the model evaluation of solute transport data is revised.

1. Characterization of peat samples: Peat soils are divers as mineral soils. Their exact characterization is crucial to classify any experimental outcome. I couldn't find any basic characterization of the investigated soil, even not the organic matter content nor the Van-Post value. Moreover, the peat is characterized as sedge peat, but Figure 1 shows moss with hyaline cells. Peat structure is a fundamental element of the study because it determines hydraulic and solute transport properties to a great extent. 2. Experimental approach: (i) The authors aim to reveal the hydraulic and solute transport properties of a fen peat. They point out that the investigated soil did not exhibit any (structure-related) bi-model behaviour, neither in the soil water retention function nor in the solute breakthrough curve. The conclusion drawn is unjustified because soil structure was compromised upon sample preparation (sieving and re-packing). Relevant and recent studies clearly showed that plant residues (embedded in an intact peat structure) in undecomposed peat may serve as a preferred solute pathway (Liu and Lennartz, 2015; Liu et al., 2016). (ii) Peat soils differ from mineral soils in their ability to retain compounds that are generally considered as "conservative" or inert such as chloride anions (Hoag and Price, 1997; Caron et al., 2015). It has been likewise observed that the application of sodium-chloride to peat samples may cause a rearrangement of pore structure based on a pore dilation effect (Ours et al., 1997). Both aspects shall be discussed in a publication on sodium-chloride transport in peat soils. 3. Model evaluation: The authors employed the mobile-immobile solute transport concept and the according solution of the underlying equations to obtain parameter values by running optimization algorithms. From a huge body of literature (for instance Parker and van Genuchten, 1984; Bond and Wierenga, 1990; Gao et al., 2009) it is evident that the simultaneous optimization of the pore water velocity (v) and the fraction of immobile water ($\beta$) may lead to invalid parameter values because both parameters account for the position of the BTC on the (dimensionless) time axis. Keeping v fixed at the measured values would probably have produced a different outcome and a different conclusion. In this context, the possible retardation of the "conservative" tracer needs to be considered (see above). In its current state the discussion of the obtained

parameter values is poor. For instance, how can it be explained that the optimized pore water velocity (v(fit)) showed greater values than the measured (v(measured)) employing the (single modal) convective dispersion equation? A ratio of v(measured)/v(fit) of less than 1 is a clear indication of a preferred solute transport situation. 4. Additional comments: There are several additional issues with the manuscript. Solute concentration data shall be presented against exchanged pore volume instead of absolute time to better classify the breakthrough behaviour. The result section starts with the presentation of the (adjusted) bulk density etc. That clearly belongs to Material and Methods. There are likewise some editorial flaws: Doubling of sentences, etc. (lines 101-104). At this stage, however, it doesn't make sense to give a complete list of minor issues because the study and manuscript have to be completely revised anyway. 5. Suggestion: There is some potential in the work. It is, for instance, interesting to compare saturated and partially saturated solute transport scenarios. It is likewise valuable to combine transient evaporation with tension disk measurements to obtain soil water retention and hydraulic conductivity information. To my knowledge, that has not been done before with peat soil samples. In any case, the experimental data base needs to be substantiate addressing the given points to draw definite conclusion. The study could be substantially upgraded if data from UNDISTURBED samples could be presented in comparison to the data from the disturbed samples. This holds true for the solute transport as well as for the hydraulic properties alike.

Cited literature Bond, W.J., and P.J. Wierenga. 1990. Immobile water during solute transport in unsaturated sand columns. Water Resour. Res. 26: 2475-2481. Caron, J., G. Létourneau, and J. Fortin. 2015. Electrical conductivity breakthrough experiment and immobile water estimation in organic substrates: is R = 1 a realistic assumption? Vadose Zone J. 14. doi:10.2136/vzj2015.01.0014. Gao, G., H. Zhan, S. Feng, G. Huang, and X. Mao. 2009. Comparison of alternative models for simulating anomalous solute transport in a large heterogeneous soil column. J Hydrol. 377:391-404. Hoag, R.S., and J.S. Price. 1997. The effects of matrix diffusion on solute transport and retardation in undisturbed peat in laboratory columns. J. Contam. Hydrol. 28:

193–205. doi:10.1016/S0169-7722(96)00085-X. Liu, H., and B. Lennartz. 2015. Visualization of flow pathways in degraded peat soils using titanium dioxide. Soil Sci. Soc. Am. J. 79: 757–765. doi:10.2136/sssaj2014.04.0153. Liu, H., Janssen, M., Lennartz, B. 2016 Changes in flow and transport patterns in fen peat following soil degradation. European Journal of Soil Science, 67 (6), pp. 763-772. DOI: 10.1111/ejss.12380 Ours, D.P., Siegel, D.I., Glaser, P.H. 1997. Chemical dilation and the dual porosity of humified bog peat. Journal of Hydrology, 196 (1-4): 348-360. Parker, J.C., and M. Th. Van Genuchten. 1984. Determining Transport Parameters from Laboratory and Field Tracer Experiments. Bull. 84-3, Va Agric. Exp. St., Blaacksburg, Va.

Additional readings Baird, A.J., and S.W. Gaffney. 2000. Solute movement in drained fen peat: a field tracer study in a Somerset (UK) wetland. Hydrol. Processes 14: 2489–2503. doi:10.1002/1099-1085(20001015)14:14<2489::AID-HYP110>3.0.CO;2-Q.

––––––––––––––––––––––––

---

## Referee Comment (RC2) · Anonymous Referee #2 · 7 Nov 2017

Simhayov et al. Saturated and unsaturated salt transport in peat from a constructed fen.

The tar sand industry of Alberta, Canada is mandated by law to restore the natural landscape they excavate to some semblance of its original state. Two of the major challenges they face to meet this ambitious goal are 1) restoring the minerotrophic fens, which are a major component of the regional landscape and 2) mitigating the transport of saline groundwater through the profiles of their restored peatlands. This manuscript by Simhayov et al. provides important new insights on solving these problems with special reference to the transport of dissolved salts through reconstructed fen peats. The authors performed a series of breakthrough experiments using laboratory permeameters packed with excavated fen peat. The experiments were run with both reactive and

non-reactive tracers and the results were interpreted with the aid of several mathematical models including the classic convection-dispersion equation (CDE), non-equilibrium transport (MIM or Mobile-Immiscible Model) and adsorption (OSA or On site Adsorption model). The CDE model provided the best fit for their data indicating that solute transport in their permeameters was dominated by convection-dispersion. They also found little evidence to support matrix diffusion from the dead or immobile pore space They interpreted this later finding in terms of the anatomical structure and degree of decomposition of the plant materials that make up their peat samples. Specifically, they note the low abundance of Sphagnum remains in their samples and the breakdown of cell walls in the sedge and moss particles. This interpretation specifically identifies the empty hyaline cells of Sphagnum and the intact cells of sedge tissues as the sites for dead pore space. While this interpretation appears faithful to their rigorous experiments and analyses I think it would also be prudent to consider other possible explanations for these result.

Could the absence of experimental evidence for matrix diffusion also be the result of artifacts introduced by their sampling plan? The authors first removed coarse plant materials (roots, twigs, leave) from the excavated peat samples and then packed the finer fraction into laboratory columns. This invasive procedure likely destroyed the original fabric of the excavated peat samples by altering the bulk density of the peat mass and geometry of the pore networks. It also could have created new macropores that dominated water transport through the columns. In addition, the excavation process itself could have greatly altered the original fabric of the peat in the donor peatlands. Because of the importance of this topic I would hope that the authors insert a paragraph providing a rationale for their method of packing their laboratory columns along with its advantages and limitations. I also hope that they could address the potential broader relationship of their study. Is it relevant just to the peat excavated by the tar sand industry (as I suspect) other excavated peat deposits elsewhere or are there also implication that can be drawn on the hydraulic properties of "pristine" undisturbed peatlands.

Overall this impressive experimental study is noteworthy for the rigor of its analysis and will set a high bar for future work in this field. But the authors could still improve the text by minor adjustments to its dense written style, particularly the occasional use of stand-alone pronouns, which are ambiguous, closer attention to syntax, and avoidance of overly complex constructions. The insertion of an extra paragraph discussing possible artifacts related to sampling procedures and potential application of this work would also be helpful

Detailed comments

Lines 9-10. This opening sentence is overly complex and confusing as written. I would advise revising the sentence by playing more careful attention to syntax and shortening the sentence. For example it can be changed to: The underlying processes governing solute transport were analyzed in peat from an experimentally constructed fen peatland by performing saturated and unsaturated solute breakthrough experiments using Na+ and Cl- as reactive and non-reactive solutes, respectively. A good rule of thumb with regard to syntax is to keep like elements of a sentence together,

Line 16: Please don't begin a sentence with a stand-alone pronoun (e.g. "This"), which in this context is ambiguous. Add the appropriate words or words after "This" to clarify the subject of this sentence.

Line 17: change "($\rightarrow$infinity)" to "(which approaches)"

Line 17: See my note about stand-alone pronouns above. It is always preferable to avoid introducing unintended ambiguity by avoiding this usage altogether.

Lines 27-28: Does this study have broader implications beyond the specific experimental setting and its selective process of constructing the peat columns in the lab? The concluding sentence of the abstract should address this question and point out the broader significance of these findings. Are these results specific to this experimental apparatus, study site, other reconstructed fens. Can the results provide new insights

on the limitations of the dual porosity model of pristine peatlands as well?

Lines 32-33. The subject of this sentence is too long. Please shorten the subject and move the verb to the beginning of the sentence.

Line 35: Change "which" to "that"

Line 40: Change "Generally, Sphagnum" to "Generally, in Sphagnum"

Lines 32-49: First two paragraphs: Some of the ideas expressed in these opening paragraphs should be reduced and incorporated into the Abstract to provide a better rationale for this study and its broader significance.

Line 64: Please avoid using (especially at the beginning of a sentence) a stand-alone pronoun. In this case the subject of this sentence is not clear.

Line 71: A dangerous assertion since any person's knowledge of the literature is always limited. For example, the authors could also cite Comas, X. and L. Slater (2004) Low-frequency electrical properties of peat Water Resources Research, Vol. 40, W12414, doi:10.1029/2004WR003534, 2004.

Line 77: Please add the appropriate word or words after the stand-alone pronoun "this" to clarify its meaning. Otherwise the subject of this sentence is ambiguous.

Line 85: The authors may want to consider the possibility that interactions between NaCl and dissolved organic matter (e.g. organic acids) may induce changes in the pore size and geometry as originally proposed by Ours et al. 1997 and supported by Comas & Slater 2005 or Levy et al. 2016?

Line 88: Change "We approach this" to "We approach this objective"

Lines 88-89: Were these experiments conducted in the field or lab? It would be a good idea to specify the later here.

Lines 109-112. I am confused. The previous text describes the peat used in this

experiment was dominated by sedge remains. However, but figure caption suggests that the peat was composed of Sphagnum moss. Please provide an explanation in this figure caption if possible.

Lines 112-125: How were these samples collected? A very brief description would be crucial since any method to secure peat samples for laboratory experiments will produce deformations to the original peat fabric, which will alter the hydraulic properties of the peat and therefore the affect results of the experiments. I realize that artifacts are unavoidable regardless of the sampling methods used but it would still be a good idea to address this issue here.

It appears that the material used in this experiment were collected from a pile of peat that was excavated from an intact fen and then dumped in a pile. It would probably be a good idea to add a sentence or two to describe how peat was collected from this excavated pile. Please remember that peat is a generic term covering a wide and heterogenous range of porous media.

Line 118: The authors should provide a rationale for adapting this unusual plan for packing peat into their permeameter columns. The approach will alter the fabric of the original in situ peat by preferentially removing coarser material and rearranging the packing and intra particle porosity of the finer-grained material.

Line 384: How can the bulk density and porosity be similar if they have different units of measure (gm/cm3 vs %)? Do you mean instead that they are statistically related?

Lines 307-310: Is this conclusion specific to soils that have high levels of Cl and other salts?

Line 342: Place the non-restrictive clause ("which is the ratio of advective vs diffusive transfer") directly after the subject it modifies (=Peclet number).

I suggest changing this sentence to: The Peclet number, "which is the ratio of advective vs diffusive transfer, was 33.9. for the fitted Cl- breakthrough data."

Line 356: change "this" to "this finding" to clarify the meaning of the stand-alone pronoun "this"

Lines 352-357: "Additionally, evidence found in the SEM scans of the peat used in this study (Fig. 1), shows that the cell walls have decayed, with only the skeleton of the cell remaining,"

I suggest clarifying this sentence since it is not clear whether the authors are specifically referring to the slender chlorophyllose cells (which I think is their intention) or the much larger hyaline cells that have rigid reinforcing structures and microfibril reinforcing structures that are resistant to decay (and will therefore retain their shape when dead). This paragraph also seems to contradict the opening description of the fine-grained peat that was packed into the lab permeameters.

Lines 354-355: Please add the appropriate citations to support the statement that "solute transport in peat has to be simulated using the MIM."

Lines 358-359: Either delete the adverb "As" at the beginning of this sentence or add an appropriate verb to complete this sentence.

Lines 362: Delete "these" to avoid a run on sentence. "and the robust estimates of v and D for the CDE, these were"

Line 401: Change" With this" to "With this approach"

Line 409: Did the authors consider chemical interactions between the ionic composition of the pore waters and organic acids as suggested by Ours et al. 1997?

Line 401: I suggest changing "With this we tested if the common assumption" to "With this approach we investigated whether the common assumption"

Line 410: Please add the appropriate word or words after "this" to clarify the subject of this sentence. I think they are referring to "this result" but I am not certain.

Line 411: Was the peat primarily composed of Sphagnum? Elsewhere in the text the

peat was described as sedge peat extracted from a fen.

Line 419-420: How can this statement ("has decayed enough to lose the cell walls but not enough to break the cell skeleton ") be true if the cell walls provide the structural basis of all plant cells. Please revise this statement both here and elsewhere in the text.

Line 421: Another possible reason could be the sampling plan, which selectively packed the finer-grained peat particles into the permeameters. This procedure altered the fabric of the original peat fabric, which was probably first altered by the extraction of an intact fen and the deposition of the peat into a spoil heap.

Lines 437-438:

"From the industry perspective, choosing and peat with dead end pores would allow a potential for significant attenuation."

Remove "and" from this sentence.

Are the authors concluding that partially decomposed peat (preferentially dominated by Sphagnum) provides the best material for optimizing solute attenuation in reconstructed peatlands?

---

## Author Response (AR1)

Dear Editor and anonymous reviewers

Below, we give the details of the minor revisions as requested by the editor's decision based on our previously submitted replies to the reviewer's comments.

We give full details of:

the reviewers comments (numbered, regular font),

our replies to the comments (in bold),

our revisions (in blue) and what we have changed in *italics*.

In detailing the revisions, we make reference to the line numbers (of the non tracked manuscript version) where we have made changes. Changes can also be seen in the tracked manuscript attached to this document.

**Revisions following Reviewer #1 comments**

 "Characterization of peat samples: Peat soils are divers as mineral soils. Their exact characterization is crucial to classify any experimental outcome. I couldn't find any basic characterization of the investigated soil, even not the organic matter content nor the Van-Post value."
 We already did give explicit reference to previously published studies describing the structure of the peat on page 4 lines 101-106 (we will add text - refer also to comment 2)).

The paragraph now on lines 105-112 reads (changes in italics): Added text to lines 101-107 detailing the composition of the peat, its origin and disturbed nature including sample collection.

"The peat used for the fen was moderately decomposed rich fen, sedge peat with remnants of Sphagnum moss, originating from a donor fen prior to stripping of overburden material to expose the oil sands deposits (Price et al., 2011; Daly et al., 2012; Nwaishi et al., 2015). The donor fen had been drained for two years prior and the peat underwent accelerated decomposition due to exposure to oxygen (Nwaishi et al., 2015). Vegetation growth on the drained fen resulted in addition of stems and leaves to the peat. The samples were shoveled into 20L buckets from a stockpile made by the heavy machinery that removed the peat layer from the donor fen, further disturbing the peat, as it was placed in the fen. The peat has a relatively open structure (Fig. 1), compared to Sphagnum peat used in other transport studies (e.g. Hoag and Price, 1989; Rezanezhad et al., 2012)."

2) "Moreover, the peat is characterized as sedge peat, but Figure 1 shows moss with hyaline cells." —Please refer to page 15 lines 348-350, where we explicitly state that: "The peat of the Nikanotee Fen watershed was moderately decomposed sedge peat containing small amounts of Sphagnum moss (Nwaishi et al., 2015)" in the Results section. It explains Figure 1 which shows how the remnant sphagnum appears in the sedge peat. With respect to comment 1+2, we will explicitly describe the peat material in the Materials and Methods earlier on in the manuscript.

Added text to lines 105-112 detailing the composition of the peat, its origin and disturbed nature including sample collection. See item 1 for full text.

3) "The conclusion drawn is unjustified because soil structure was compromised upon sample preparation (sieving and re-packing). Relevant and recent studies clearly showed that plant residues (embedded in an intact peat structure) in undecomposed peat may serve as a preferred solute pathway (Liu and Lennartz, 2015; Liu et al., 2016)."

—We disagree. The comment correctly identifies, that we reconstituted the peat. However, the field site is characterized as a recently replaced and reconstituted site. In this respect, the laboratory protocol is similar to what has happened in the field, since undisturbed samples (i.e. naturally grown peat) does not exist in our system (See lines 102-105). More specifically, the peat had been disturbed a few years prior to sampling. It had been taken from a pile that was created using heavy machinery. We think it is very likely that these disturbances had a much greater effect on the peat properties than the lab preparation. The preparation was designed to reduce the variability caused by the disturbances. Furthermore, nowhere in the text is "sieving" mentioned. The peat was not sieved, as suggested by reviewer 1, but coarse material (leaves and twigs and stem from the aforementioned disturbances which accounted to about 2% in volume) was carefully removed and the peat was gently mixed to increase homogeneity. We don't believe this caused a change to the cellular structure of the cells. However, we do agree that we perhaps did not make it sufficiently clear to the reader that our samples are from a stockpile of heavily disturbed peat, which was replaced in the newly created artificial peatland. Thus, we will be careful to address the full characterization of the peat (also cf. comment 1) and 2) in the revised manuscript.

**Lines 87-89: Added sentence: "Additionally, acknowledging that Na + and Cl- ions may interact with dissolved organic matter, inducing changes in the pore size and geometry (Ours et al., 1997; Comas & Slater, 2004) a pre-treatment was implemented (see subsection 2.3.1 - Sample preparation)."**

4) "Peat soils differ from mineral soils in their ability to retain compounds that are generally considered as "conservative" or inert such as chloride anions (Hoag and Price, 1997; Caron et al., 2015). It has been likewise observed that the application of sodium-chloride to peat samples may cause a rearrangement of pore structure based on a pore dilation effect (Ours et al., 1997). Both aspects shall be discussed in a publication on sodium-chloride transport in peat soils"

—We agree with the reviewers' comment, and in recognizing the potential for sodium to cause flocculation and/or pore dilation, we flushed the column with NaCl as a pretreatment of the samples to minimize the effect during experimentation. We stated this explicitly on lines 159-162 and a lines 87-89.

**No changes required.**

5) "Model evaluation: The authors employed the mobile-immobile solute transport concept and the according solution of the underlying equations to obtain parameter values by running optimization algorithms. From a huge body of literature (for instance Parker and van Genuchten, 1984; Bond and Wierenga, 1990; Gao et al., 2009) it is evident that the simultaneous optimization of the pore water velocity (v) and the fraction of immobile water ( $\beta$ ) may lead to invalid parameter values because both parameters account for the position of the BTC on the (dimensionless) time axis. Keeping v fixed at the measured values would probably have produced a different outcome and a different conclusion."

—Of course, fixing parameters will lead to different conclusions, we are aware of this. The method the reviewer proposes is for exactly that same reason a very unsatisfying method. Pre-fixing either the pore water velocity (v) or the fraction of immobile water (β) is subjective. Moreover, if parameters are fixed a priori, the inverse modeling will not provide evidence on the flow phenomena, but merely "prove" the conceptual model which was used to fix the initial parameters in the first place. For this reason, we took a different approach; we refrained from fixing parameters, and used the information content of the experiments. Thus, we attest on whether or not the mobile-immobile solute transport model is an over-parameterization or an adequate model choice. Moreover, simultaneously fitting v and β is possible and has been done previously (e.g. Zurmühl, 1998; Tang et al., 2009 and references herein). Considering the implied 1:1 representation of v for β (or vice versa) close inspection of inspection of Eq. 1 shows that what the reviewer implies cannot be the case. Division by β leads to β being part of each of the three summands on the right-hand side of the Eq. 1, and therefore contradicts the reviewer's assumption. We believe our approach is appropriate.

**No changes needed**

6) "In this context, the possible retardation of the "conservative" tracer needs to be considered (see above).

— We disagree that a potential Cl adsorption might be relevant on the scale of the flow through reactors. There is a theoretical potential, that an anion such as Cl- (or Br- used in other studies) might be attenuated, too, but this would contradict what is known about Cl- in peat. Since no studies exists (to the knowledge of the authors) of deuterium as tracer in peat, it is very speculative to state that Cl should not be at least very close to conservative as a tracer. Hoag and Price (1997) indicated Cl was subject to physical diffusion into inactive pores as evidenced by the Cl retardation factor greater than 1 (R=1.1). The reviewer is commenting on the classical definition of retardation (by adsorption). We acknowledge that while most studies treat Cl as a conservative tracer, there is a theoretical possibility that a small anion adsorption effect may exist, which can only be distinguished by known conservative tracers like deuterium. We will add text acknowledging the possibility.

After discussion of parameter  $\beta$  we now additionally turn to  $\omega$ : We now argue by adding: with the following sentence in line 356-358: "*This exclusion is supported by the instantaneous equilibration between the mobile and the immobile zone, as indicated by the very large*  $\omega$  *which was at the upper bound during the parameter estimation*".

Table 3 now gives the estimated  $\omega$  values, and we corrected the  $\alpha_{MiM}$  of Cl to 0.9 (min-1), too. By doing so, we now indicate in the table and the caption, that for the Cl break through curve using the MiM model, the coefficient of variation cannot be estimated using the classical first-order second moment approach: "*ne* – *not evaluated: the parameter is at the upper feasible bound of the parameter estimation. (Toride et al. 1995)*"

Finally, the introduction of omega required that we define it mathematically, which we now do in a footnote to Table 9:  $\omega_{MIM} = \alpha * L / (\theta * v)$ ,  $\omega_{OSA} = (\alpha (R - 1) * L) / v$

7) "For instance, how can it be explained that the optimized pore water velocity (v(fit)) showed greater values than the measured (v(measured)) employing the (single modal) convective dispersion equation? A ratio of v(measured)/v(fit) of less than 1 is a clear indication of a preferred solute transport situation."

--We think the reviewer refers to the effective pore water velocity. Of course, the comparison of v(fit) with v(measured) can be done, but differences are to be expected from normal measurement errors, experimental set up, and remaining air bubbles in the sample. In any case,  $\omega$  [-], the dimensionless mass-transfer coefficient also indicated at the non-existence of mobile-immobile solute transport as it was found that  $\omega > 100$  (In the paper we actually reported on the dimensioned mass transfer coefficient,  $\alpha$ ). We realize that our conclusions from our results of  $\omega$  were not presented with sufficient clarity. The parameter  $\omega$  could also not be identified by inverse modeling. In the inversion,  $\omega$  ran into the maximum upper boundary limit of 100 in CXTFIT. This supports the conclusion, that the MIM solute transport model is an over-parameterization (corroborated by the CXTFIT results files, too), and that the CDE is sufficient for describing the observed breakthrough and solute transport. For this reason, we are convinced that our results and conclusions are not affected by this comment. In a revised manuscript, we will address this by including a brief discussion on  $\omega$ , and present the values in Table 3, and hope this can strengthen the paper considerably.

**No changes needed**

8) "Solute concentration data shall be presented against exchanged pore volume instead of absolute time to better classify the breakthrough behavior."

---Both methods of presentation can be found in the literature (e.g. Tang et al., 2009). Moreover, we took great care in presenting and discussing our results indicating CDE solute transport, and therefore do not believe it will change the understanding of the chart or the processes. If the editor finds this change compulsory, we could consider this in a revised manuscript.

**No changes needed**

9) "The result section starts with the presentation of the (adjusted) bulk density etc. That clearly belongs to Material and Methods. "

- A good point. While being more elaborate on the description of the peat, (cf. comment 1-3), we will address this, too, and make the Material and Methods more concise and complete.

We rephrased the sentence now in line 291-292, to clarify the bulk density results, which now reds: "*The bulk density and porosity of the prepared peat samples in the various experiments was similar (Table 1) indicating a successful sample replication.*"

10) "There are likewise some editorial flaws: Doubling of sentences, etc. (lines101-104)."
 —We apologize for this and will make an even greater effort in addressing any potential editorial concerns, thoroughly, before submitting a revised manuscript.

We thoroughly checked and corrected the document for editorial flaws which might have slipped our attention before submission. We have now deleted the repeated sentence.

11) "Suggestion: There is some potential in the work. It is, for instance, interesting to compare saturated and partially saturated solute transport scenarios. It is likewise valuable to combine transient evaporation with tension disk measurements to obtain soil water retention and hydraulic conductivity information."

—We agree this is interesting, but do not believe it is central to the manuscript; it would disrupt the flow, and in its extent is a separate study for another time.

**No changes needed**

12) "In any case, the experimental data base needs to be substantiate addressing the given points to draw definite conclusion. The study could be substantially upgraded if data from UNDISTURBED samples could be presented in comparison to the data from the disturbed samples. This holds true for the solute transport as well as for the hydraulic properties alike".

—We agree that comparison to undisturbed samples is always interesting; however, the peatland from which these samples were drawn was destroyed long before we began this experiment. So in that sense, we disagree, as undisturbed samples do not exist. The samples are obtained from stockpiled and heavily disturbed original material (during placement on site). Our focus here is to characterize the disturbed peat, so the implications for solute transport at the specific site can be discussed. In a revised manuscript, we will include a clear statement on the objectives, and, again, be much more precise on the nature of our samples.

Added text to lines 105-112 detailing the composition of the peat, its origin and disturbed nature including sample collection. See item 1 for full text.

References used in Reply to Reviewer #1:

Hoag, R.S., and Price, J.S. (1997). The effects of matrix diffusion on solute transport and retardation in undisturbed peat in laboratory columns. Journal of Contaminant Hydrology 28:193–205.

Nwaishi, F., Petrone, R. M., Price, J. S., Ketcheson, S. J., Slawson, R., & Andersen, R. (2015). Impacts of donor-peat management practices on the functional characteristics of a constructed fen. Ecol. Eng., 81, 471-480.

Tang, G., Mayes, M. A., Parker, J. C., Yin, X. L., Watson, D. B., and Jardine, P. M. (2009). Improving parameter estimation for column experiments by multi-model evaluation and comparison, Journal of Hydrology 376, 567–578.

Zurmühl, T. (1998). Capability of convection–dispersion transport models to predict transient water and solute movement in undisturbed soil columns, Journal of Contaminant Hydrology, 30, 101–128.

**Revisions following Reviewer #2 comments**

1. "Lines 9-10. This opening sentence is overly complex and confusing as written. I would advise revising the sentence by playing more careful attention to syntax and shortening the sentence. For example it can be changed to: The underlying processes governing solute transport were analyzed in peat from an experimentally constructed fen peatland by performing saturated and unsaturated solute breakthrough experiments using Na+ and Cl- as reactive and non-reactive solutes, respectively. A good rule of thumb with regard to syntax is to keep like elements of a sentence together".
—We agree and appreciate this helpful comment. In our revised manuscript, we will be careful in enhancing readability, e.g. by adopting proposed sentence.

Changed lines 11-13 to: "The underlying processes governing solute transport were analyzed in peat from an experimentally constructed fen peatland by performing saturated and unsaturated solute breakthrough experiments using Na+ and Cl- as reactive and non-reactive solutes, respectively."

2. "Line 16: Please don't begin a sentence with a stand-alone pronoun (e.g. "This"), which in this context is ambiguous. Add words or words after "This" to clarify the subject of this sentence."
—we will try to address this throughout the document.

Now line 18: We changed the instances of stand-alone pronouns to be grammatically correct. Corrected for the entire document.

"Line 17: change "(!infinity)" to "(which approaches)""
we will change this as proposed.

Changed lines 18-20 to: "Furthermore, the very high Damköhler number (which approaches infinity) found suggests instantaneous equilibration between the mobile and immobile phases; underscoring the redundancy of the MIM approach for this particular peat."

4. "Line 17: See my note about stand-alone pronouns above. It is always preferable to avoid introducing unintended ambiguity by avoiding this usage altogether."
—We agree and will change it to be decisive.

Now line 18, we changed to: "(->infinity)" to "(which approaches infinity)".

5. "Lines 27-28: Does this study have broader implications beyond the specific experimental setting and its selective process of constructing the peat columns in the lab? The concluding sentence of the abstract should address this question and point out the broader significance of these findings. Are these results specific to this experimental apparatus, study site, other reconstructed fens. Can the results provide new insights on the limitations of the dual porosity model of pristine peatlands as well?"

— We agree with both comments/questions. The study does have a broader implication, and we think we can exploit this aspect more than we have done. We will state the significance of our study/conclusions in the abstract of a revised manuscript. i.e. the mobile-immobile solute transport model cannot be selected per se.

Added to lines 29-30 and lines 455-456: "... and imply that MIM should not be automatically assumed for solute transport in peat but should rather be evidence based."

*6.* "Lines 32-33. The subject of this sentence is too long. Please shorten the subject and move the verb to the beginning of the sentence. "

---As stated previously, we will be more specific in the use of our language and address this in a revised manuscript.

Now lines 34-35: Shortened the subject and moved the verb to the beginning of the sentence.

7. "Line 35: Change "which" to "that""—see for previous comment.

Line 39: Changed "which" to "that".

\*Line 40: Change "Generally, Sphagnum" to "Generally, in Sphagnum"
—as previous comment.

Line 40: Changed "Generally, Sphagnum" to "Generally, in Sphagnum".

9. "Lines 32-49: First two paragraphs: Some of the ideas expressed in these opening paragraphs should be reduced and incorporated into the Abstract to provide a better rationale for this study and its broader significance."

---We understand this is related to the comments on lines 27-28, and will, in a revised manuscript, carefully reassess how to make the abstract stronger. This is a good suggestion, it is logical and helps strengthen the essence of the article in the abstract, and will be addressed in the revised manuscript.

Now lines 34-49. Rewritten to be more concise.

10. "Line 64: Please avoid using (especially at the beginning of a sentence) a stand-alone pronoun. In this case the subject of this sentence is not clear."

---we agree and will check our manuscript for these instances.

Line 64: corrected standalone pronoun.

"Line 71: A dangerous assertion since any person's knowledge of the literature is always limited.
For example, the authors could also cite Comas, X. and L. Slater (2004) Low frequency electrical properties of peat Water Resources Research, Vol. 40, W12414, doi:10.1029/2004WR003534, 2004."
—We agree, and in the following, we will be more reserved in the revised manuscript by writing: "As to the current knowledge of the authors.". Additionally, we will consider the mentioned publication, and possible implications from it. We also added the suggested citation.

**Now line 70: "As to the knowledge of the authors..". We also added the suggested citation.**

12. "Line 77: Please add the appropriate word or words after the stand-alone pronoun "this" to clarify its meaning. Otherwise the subject of this sentence is ambiguous. "
—We will replace 'this' by stating "[...], the method of estimating ne from photo-imagery may easily lead to a systematic miscalculation of effective pore water velocity."

Line 77: We replaced 'this' with the sentence: "The method of estimating  $n_e$  from photo-imagery may easily lead to a systematic miscalculation of effective pore water velocity."

13. "Line 85: The authors may want to consider the possibility that interactions between NaCl and dissolved organic matter (e.g. organic acids) may induce changes in the pore size and geometry as originally proposed by Ours et al. 1997 and supported by Comas & Slater 2005 or Levy et al. 2016? "
—We fully agree with this statement, and had, for this reason, explicitly included a pretreatment to equilibrate and flush the samples. This is clearly stated in lines 152-155 and see no necessity to for changes.

Added sentence in line 87: "Additionally, acknowledging that Na + and Cl- ions may interact with dissolved organic matter, inducing changes in the pore size and geometry (Ours et al., 1997; Comas & Slater, 2004) a pre-treatment was implemented (see subsection 2.3.1 - Sample preparation)."

**14.** "Line 88: Change "We approach this" to "We approach this objective"" —we will correct this as suggested.

Changed "We approach this" to "We approach this objective" now in line 92

15. "Lines 88-89: Were these experiments conducted in the field or lab? It would be a good idea to specify the later here."

—we are surprised that this was not clear, but will add the following sentence in the revised manuscript: "lab based experiments including" in line 88.

We added a "We approach this objective by conducting lab based experiments including saturated and unsaturated breakthrough experiments using NaCl.", now in line 92-93.

**16.** "Lines 109-112. I am confused. The previous text describes the peat used in this experiment was dominated by sedge remains. However, but figure caption suggests that the peat was composed of Sphagnum moss. Please provide an explanation in this figure caption if possible. "

—We realize we did not elaborate on this point sufficiently, and failed to make it clear to the reader, since also the other reviewer commented on this. Therefore, we will be more specific on the description of the peat characteristics (e.g. by repeating the findings of other researchers in a revised manuscript). We do note, that we give references to other research on the same constructed peatland, and we do present a scanning electron microscopy images of the Sphagnum remnants in the peat samples (lines 348-350).

Added text to now in lines 105-112 detailing the composition of the peat, its origin and disturbed nature including sample collection. See item 1 in answer to reviewer #1 for full text.

17. "Lines 112-125: How were these samples collected? A very brief description would be crucial since any method to secure peat samples for laboratory experiments will produce deformations to the original peat fabric, which will alter the hydraulic properties of the peat and therefore the affect results of the experiments. I realize that artifacts are unavoidable regardless of the sampling methods used but it would still be a good idea to address this issue here. It appears that the material used in this experiment were collected from a pile of peat that was excavated from an intact fen and then dumped in a pile. It would probably be a good idea to add a sentence or two to describe how peat was collected from this excavated pile. Please remember that peat is a generic term covering a wide and heterogenous range of porous media. "

—This concern was also raised by the other reviewer, and we seem to have failed to be precise in describing the a) characteristics of the peat, b) the sampling method, and c) the site description. As we have mentioned before, we will be careful to present the information more coherently in a revised manuscript. We do point out, that at several points in the manuscript we give descriptions, e.g. lines 102-105 where we state: " the peat has been disturbed for a few years prior to sampling. It was taken from a pile that was created using heavy machinery. It is therefore likely that these disturbances had a much greater effect on the peat properties than the lab preparation. Further the preparation was designed to reduce the variability caused by the disturbances.". Moreover, the peat was not sieved, as suggested by reviewer 1, but coarse material was carefully removed (<=2% in volume) and the peat was gently mixed to increase homogeneity. It is unlikely this caused a change to the cellular structure of the cells.

**See item 16 above.**

18. "Line 118: The authors should provide a rationale for adapting this unusual plan for packing peat into their permeameter columns. The approach will alter the fabric of the original in situ peat by preferentially removing coarser material and rearranging the packing and intra particle porosity of the finer-grained material. "

-See answer to previous comment.

Provided a rationale for sample homogenization during sample preparation as an addition to the lines 122-125 "As previously noted, the peat was sampled from the stock of disturbed peat used to construct the fen; in addition, we carefully removed woody inclusions and intact leaves to homogenize it such that we could ensure minimal variation between samples. The peat was gently, yet thoroughly mixed and packed into columns (see appendix A.1); no milling or sieving was done." Also see lines 105-112 regarding sample collection.

19. "Line 384: How can the bulk density and porosity be similar if they have different units of measure (gm/cm3 vs %)?"

— We think the reviewer refers to line 284 (not 384): This similarity is not apparent, in Table 1 the bulk density is clearly given by 0.12 g/cm3 and porosity by 0.93 [-]. The numbers with % sign are the coefficient of variation as mentioned in the caption, which is the standard deviation of the three replicates divided by their mean. Therefore, we do not believe we need to change anything here. "Do you mean instead that they are statistically related? "

---No, it is a measure for the similarity of the replicates.

**No changes needed.**

20. "Lines 307-310: Is this conclusion specific to soils that have high levels of Cl and other salts?" —It is specific to the use of EC electrodes to monitor the transition of reactive ions in any solution that flows through a reactive medium. We will incorporate this explanation.

Now on Lines 315-319: Added an explanation for the use of EC electrodes to monitor reactive solute flow through a reactive medium. "The dissimilarity of the EC breakthrough curve to that of Na+ (Fig. 4) demonstrates the limitation of using EC electrodes as an indicator for solutions containing reactive solutes, flowing through reactive mediums. This limitation is due to enrichment of ions in the solution from the soil and cation exchange with the medium, which changes the solution concentration of the cation of interest; therefore, EC can be a good estimator for non-reactive solutes but is limited as an indicator for cation transport (Olsen et al., 2000; Vogeler et al., 2000)."

**21.** "Line 342: Place the non-restrictive clause ("which is the ratio of advective vs diffusive transfer") directly after the subject it modifies (=Peclet number). I suggest changing this sentence to: The Peclet number, "which is the ratio of advective vs diffusive transfer, was 33.9. for the fitted CI- breakthrough data." "

---Agreed. This is a sensible suggestion and will make the changes accordingly in the revised manuscript. In the next comments, we will limit our response to "agreed", indicating we will revise the manuscript accordingly.

Now in line 351: changed sentence to: "The Peclet number, *which is the ratio of advective vs diffusive transfer*, was 33.9 for the fitted Cl- breakthrough data."

22. "Line 356: change "this" to "this finding" to clarify the meaning of the stand-alone pronoun "this""

**--- agreed.**

Now in line 358 we changed "this" to "this finding"

23. "Lines 352-357: "Additionally, evidence found in the SEM scans of the peat used in this study (Fig. 1), shows that the cell walls have decayed, with only the skeleton of the cell remaining," I suggest clarifying this sentence since it is not clear whether the authors are specifically referring to the slender chlorophyllose cells (which I think is their intention) or the much larger hylaline cells that have rigid reinforcing structures and microfibril reinforcing structures that are resistant to decay (and will therefore retain their shape when dead). This paragraph also seems to contradict the opening description of the fine-grained peat that was packed into the lab permeameters. "

— We thank you for this helpful and knowledgeable sentence. In discussing the decomposition of the Sphagnum moss parts, we were not specific enough. Generally, the sentences (lines 352-357) relate to the decomposition of the membranes of the hyaline cells. In the presence of intact hyaline cells, the cells act as dead-end pores, where the membranes restrict the solute transport to very small openings. Once the hyaline cell membranes have decayed, solutes can be transported by advection through-out the inner-plant matrix. Thus, the dead end pores no longer exist, a fact which can serve as an explanation for why the parameterization of solute transport models does not indicate a mobile and an immobile domain. While the reviewers comment does not contradict our findings, we will elaborate carefully on the different structures in the peat in a revised manuscript. Finally, the comment does not contradict our study description and methods, since it is unlikely that the careful packing impacted cell membranes (as discussed in the answer to comment on lines 112-115).

**Now in lines 363-365 we changed the expression "cell walls" to "cell membrane".**

24. "Lines 354-355: Please add the appropriate citations to support the statement that "solute transport in peat has to be simulated using the MIM.""

—We will gladly address this by giving more references to the literature, by rephrasing and providing references: 'These results contradict the hitherto common finding in laboratory studies that breakthrough experiments on peat need to be described by the MIM (Hoag and Price, 1997; Liu et al., 2016; Rezanezhad et al., 2012; Rezanezhad et al., 2017; Thiemeyer et al., 2017)'

Now in line 365-369 we rephrased the section and provided references: "*These results contradict the hitherto common finding in laboratory studies that breakthrough experiments on peat need to be described by the MIM (Hoag and Price, 1997; Rezanezhad et al., 2012; Liu et al., 2016; Rezanezhad et al., 2017; Thiemeyer et al., 2017)*".

25. "Lines 358-359: Either delete the adverb "As" at the beginning of this sentence or add an appropriate verb to complete this sentence."

---Agreed.

**Changed "As" to "Since".now in lines 370-371.**

**26. "Lines 362: Delete "these" to avoid a run on sentence. "and the robust estimates of v and D for the CDE, these were""**

---Agreed.

Deleted "these" and rephrased the sentence, now in line 373-374: "Having shown that the MIM is not parsimonious in its parameters, the robust estimates of v and D for the CDE were fixed when fitting the remaining model parameters of the CDE and one-site adsorption model for Na+."

27. "Line 401: Change" With this" to "With this approach"" — *Agreed.*

We changed "this" to "*this approach*", now in line 414.

*28.* "Line 409: Did the authors consider chemical interactions between the ionic composition of the pore waters and organic acids as suggested by Ours et al. 1997? " *—See answer to comment on line 85.*

See answer to item 13.

29. "Line 401: I suggest changing "With this we tested if the common assumption" to "With this approach we investigated whether the common assumption""
—Agreed.

We changed "With this we tested if the common assumption [...] " to "*With this approach we investigated whether the common assumption* [...]" now in line 414.

30. "Line 410: Please add the appropriate word or words after "this" to clarify the subject of this sentence. I think they are referring to "this result" but I am not certain."
—Agreed.

We changed "this" to "*this result*", now in line 423.

31. "Line 411: Was the peat primarily composed of Sphagnum? Elsewhere in the text the peat was described as sedge peat extracted from a fen."
—See lines 348-350.

See answer on item 16.

**32.** "Line 419-420: How can this statement ("has decayed enough to lose the cell walls but not enough to break the cell skeleton ") be true if the cell walls provide the structural basis of all plant cells. Please revise this statement both here and elsewhere in the text. "

--- We inadvertently used the word "cell walls" while we meant "cell membranes". We will correct this in a revised manuscript.

We changed "cell walls" to "cell membranes", now in line 432-433:

**33.** "Line 421: Another possible reason could be the sampling plan, which selectively packed the finer-grained peat particles into the permeameters. This procedure altered the fabric of the original peat fabric, which was probably first altered by the extraction of an intact fen and the deposition of the peat into a spoil heap. "

— There is no dispute that any manipulation of a sample alters it. However, the prior disturbance, as mentioned in the comment, is likely to have had a greater impact. The peat was not manipulated in a destructive manner such as sieving or milling. The reduced variability as a result of the careful homogenization process provides the improved ability to understand the hydraulic properties of the peat. While we doubt the sample preparation had a notable effect on the value of the parameters, we are certain it did not invalidate our interpretation of the processes, and thus our conclusions. As indicated elsewhere in this reply and in the reply to the reviewer #1, we will make it clearer in a revised manuscript, that both the sampling methods and sampling treatments, in the light of the type of sampling site, are acceptable.

Now starting on line 434, we added a paragraph stating: "Although we acknowledge that any manipulation of a sample alters it, the prior disturbance (see introduction of the method section) is likely to have had a significantly greater impact. Furthermore, the peat was not manipulated in a destructive manner such as sieving or milling and handled carefully. The reduced variability as a result of the careful homogenization process provides the improved ability to understand the hydraulic properties of the peat.

34. "Lines 437-438: "From the industry perspective, choosing and peat with dead end pores would allow a potential for significant attenuation." Remove "and" from this sentence."
— Agreed.

Now in line 454-455 we removed the "and" from this sentence.

**35.** "Are the authors concluding that partially decomposed peat (preferentially dominated by Sphagnum) provides the best material for optimizing solute attenuation in reconstructed peatlands?" —The sentence tries to convey that the hydraulic and transport properties of the peat should be checked and match the desired function. Therefore, to answer the reviewers question, if solute attenuation is the goal then peat with a larger amount of sphagnum and a dual porosity structure would be a better choice. We will add additional text in the revised manuscript.

Now in lines 454-455 we changed sentence to "From the industry perspective, if solute attenuation is the goal then peat with a larger amount of sphagnum and a confirmed dual porosity structure would be a better choice".

**References used in Reply to Reviewer #2:**

Hoag, R.S., and Price, J.S. (1997). The effects of matrix diffusion on solute transport and retardation in undisturbed peat in laboratory columns. Journal of Contaminant Hydrology 28:193–205.

Liu, H., Janssen, M., and Lennartz, B. (2016). Changes in flow and transport patterns in fen peat following soil degradation. European Journal of Soil Science 67:763–772.

Rezanezhad, F., Price, J.S., & Craig, J.R. (2012). The effect of dual-porosity on transport and retardation in peat: A laboratory experiment. Can. J. Soil Sci., 92: 1-10.

Rezanezhad, F., Kleimeier, C., Milojevic, T., Liu, H., Weber, T.K.D., Van Cappellen, P., and Lennartz, B. (2017). The Role of Pore Structure on Nitrate Reduction in Peat Soil: A Physical Characterization of Pore Distribution and Solute Transport, Wetlands, doi 10.1007/s13157-017-0930-4.

Tiemeyer, B., Pfaffner, N., Frank, S., Kaiser, K., Fiedler, S., (2017). Pore water velocity and ionic strength effects on DOC release from peat-sand mixtures: results from laboratory and field experiments. Geoderma 296:86-97.

**1 Saturated and unsaturated salt transport in peat from a constructed**

**2 **fen**

3 Reuven B. Simhayov1, Tobias K. D. Weber1,2,\*, Jonathan S. Price1

4 1Department of Geography, University of Waterloo, Waterloo, Ontario, N2L 3G1, Canada

5 2Soil Science and Soil Physics Division, Institute of Geoecology, TU Braunschweig, Langer Kamp 19c, 38106
 6 Braunschweig, Germany

7

\* now at: Institute for Soil Science and Land Evaluation, Biogeophysics, University of Hohenheim, Emil-Wolff-Straße 27,
9 DE-70599 Stuttgart, Germany. Email: tobias.weber@hohenheim.de

10 Correspondence to: Reuven B. Simhayov (rbsimhay@uwaterloo.ca)

11 Abstract. The underlying processes governing solute transport were analyzed in peat from an experimentally constructed fen

12 peatland were analyzed by performing saturated and unsaturated solute breakthrough experiments using Na+ and Cl- as

13 reactive and non-reactive solutes, respectively. To determine the underlying processes of solute transport in peat an

14 experimental constructed fen peatland, soil hydraulic properties were measured and saturated and unsaturated solute

15 breakthrough experiments were performed using Na+ and Cl- as reactive and non-reactive solutes, respectively. We tested the

16 performance of three solute transport models, including the classical equilibrium Convection-Dispersion Equation (CDE), a

17 chemical non equilibrium one-site adsorption model (OSA) and a model to account for physical non-equilibrium, the

18 mobile-immobile phases (MIM). The selection was motivated by the fact that the applicability of the MIM in peat soils finds

19 a wide consensus. However, results from inverse modelling and a robust statistical evaluation of this peat provide evidence

20 that the measured breakthrough of the conservative tracer, Cl- could be simulated well using the CDE. This is demonstrated

21 by aFurthermore, the very high Damköhler number (which approaches  $\rightarrow$  infinity) found suggesting suggests instantaneous

22 equilibration between the mobile and immobile phases; this underscores underscoring the redundancy of the MIM approach

[revised manuscript text omitted]